# Prolyl isomerization controls activation kinetics of a cyclic nucleotide-gated ion channel

Philipp A. M. Schmidpeter [1], Jan Rheinberger [1,3] & Crina M. Nimigean [1,2✉]

SthK, a cyclic nucleotide-modulated ion channel from *Spirochaeta thermophila*, activates slowly upon cAMP increase. This is reminiscent of the slow, cAMP-induced activation reported for the hyperpolarization-activated and cyclic nucleotide-gated channel HCN2 in the family of so-called pacemaker channels. Here, we investigate slow cAMP-induced activation in purified SthK channels using stopped-flow assays, mutagenesis, enzymatic catalysis and inhibition assays revealing that the *cis/trans* conformation of a conserved proline in the cyclic nucleotide-binding domain determines the activation kinetics of SthK. We propose that SthK exists in two forms: *trans* Pro300 SthK with high ligand binding affinity and fast activation, and *cis* Pro300 SthK with low affinity and slow activation. Following channel activation, the *cis/trans* equilibrium, catalyzed by prolyl isomerases, is shifted towards *trans*, while steady-state channel activity is unaffected. Our results reveal prolyl isomerization as a regulatory mechanism for SthK, and potentially eukaryotic HCN channels. This mechanism could contribute to electrical rhythmicity in cells.

[1] Weill Cornell Medicine, Department of Anesthesiology, 1300 York Avenue, New York, NY 10065, USA. [2] Weill Cornell Medicine, Department of Physiology and Biophysics, 1300 York Avenue, New York, NY 10065, USA. [3] Present address: University of Groningen, Groningen, Netherlands. ✉email: crn2002@med.cornell.edu

on channels involved in processes such as electrical signaling across the cell membrane display tightly regulated activation. For example, the generation of an action potential requires that voltage-gated $Na^+$ channels open fast to allow $Na^+$ influx into the cell in order to rapidly depolarize the membrane[1–3]. Slower-activating voltage-gated $K^+$ channels are required in order to allow $K^+$ efflux and re-polarize, and, in some cases, hyper-polarize the membrane[2,4]. In some cells, hyperpolarization activates hyperpolarization-activated channels (HCN) that reset the resting potential and thereby enable the next action potential[5,6].

In the heart, the expression of HCN channels is high in the sinoatrial node and the activity of HCN2 and HCN4 channels contributes to the autonomous rhythmicity of the heartbeat[7], and thus they are also named pacemaker channels. Besides their activation by hyperpolarization, the activity of HCN channels is further modulated by cyclic nucleotides like cAMP[8]. For HCN2, the rise in current following cAMP increase is slow and can take tens of seconds[9]. The molecular underpinnings responsible for these slow kinetics are elusive.

Here we use the bacterial channel SthK[10] to investigate the mechanism underlying the slow activation by cAMP. SthK has functional similarity to eukaryotic cyclic nucleotide-modulated channels making it a *bona fide* model system to study specific mechanistic aspects of these channels[11–15]. Importantly for this study, we showed previously that upon cAMP exposure, SthK reaches maximum channel activity slowly and in a bi-phasic manner with time constants of 20 ms and 2 s, respectively[11]. The slow phase is reminiscent of the cAMP-mediated slow increase in current in HCN2 channels[9]. In contrast, cGMP-mediated current increase in CNG channels is fast, on the order of milliseconds[16].

SthK also has the same architecture and similar sequence and structure as eukaryotic HCN and CNG channels, but unlike its eukaryotic counterparts, pure SthK protein can be easily produced in large enough quantities for structural and functional work. SthK has a voltage sensing domain, a pore domain, and a cytosolic cyclic nucleotide-binding domain (CNBD) that is connected to the pore via a helical C-linker (Fig. 1a, Supplementary Fig. 1a)[13,17,18].

Here we use mutational and enzymatic studies to show that activation kinetics of SthK channels are controlled by a proline residue in the CNBD (Pro300), conserved in all HCN but not CNG channels, that undergoes peptidyl-prolyl *cis/trans* isomerization (prolyl isomerization). The slow activation phase in SthK is abolished by either replacing Pro300 with another amino acid or by catalyzing the *cis/trans* re-equilibration with prolyl isomerases. We propose a model where pre-existing *cis/trans* heterogeneity at Pro300 in the apo state of SthK leads to bi-phasic activation kinetics as the two channel species, containing either *cis* or *trans* proline, activate with intrinsically different kinetics. Structural data and kinetic simulations further support this model.

## Results

**Pro300 determines the activation kinetics of SthK.** We previously showed that activation of SthK is slow and takes ~2 s to reach a maximum[11]. In all cyclic nucleotide-gated channels the connection between the C-linker and the CNBD is formed by a helix-loop-helix motif called the siphon[19]. Interestingly, a proline is located at the end of the first siphon helix (Pro300 in SthK), which is conserved in HCN and SthK, but not CNG, channels (Supplementary Fig. 1a)[10,13]. This proline had not been previously implicated in prolyl isomerization in HCN channels or SthK. Due to steric restrictions and the lack of an amide-proton, prolines are only rarely found at the C-termini of α-helices[20,21]. To test whether this unusual proline (Pro300, Fig. 1a, b) plays a

role in channel activation, we employed a fluorescence-based stopped-flow assay to measure activation kinetics of a SthK mutant where the proline was substituted by an alanine. In this assay, SthK is reconstituted into proteo-liposomes encapsulating ANTS fluorophore (Supplementary Fig. 1b–e) in the absence of cAMP to silence channels that are oriented with their CNBDs towards the inside of the vesicles. In a first mixing step liposomes are exposed to cAMP to activate SthK channels (only channels with their CNBDs facing outwards respond to cAMP application). In a second mixing step, $Tl^+$ is added and the rate of ANTS quenching caused by $Tl^+$-entry through open channels is measured at defined delay times after cAMP application (12 ms–10 s, Fig. 1d). In contrast to the bi-phasic and slow activation kinetics observed with WT SthK, activation of SthK P300A is fast, and complete after the shortest mixing time (12 ms, Fig. 1e, f). On the other hand, SthK P300A shows similar single-channel amplitudes and open probabilities to WT SthK, suggesting that only the activation time course was affected by the mutation (Fig. 1f and Supplementary Fig. 2a–c). Furthermore, neither WT SthK nor SthK P300A showed measurable activity in the absence of cAMP (gray lines in Fig. 1d, e), similar to the quenching observed for protein-free liposomes (Supplementary Fig. 1d).

The covalent linkage between the sidechain and the backbone in proline 1) increases the main chain rigidity and 2) strongly increases the likelihood for adopting *cis* conformations of the peptide bond before the proline (Fig. 1c)[22]. To test whether the change in activation kinetics observed in SthK P300A was due to an increase in backbone flexibility, we also substituted Pro300 with valine. β-branched amino acids such as valine reduce the backbone flexibility to a similar extent as proline[23]. Like SthK P300A, the P300V variant also displays fast activation within milliseconds (Supplementary Fig. 2d, e), arguing against a rigidifying effect of Pro300 in SthK.

**Prolyl isomerases accelerate the activation time course.** We next investigated whether the possibility for Pro300 to adopt two distinct configurations, *cis* or *trans*, is responsible for the slow activation in SthK. Within a working model where apo SthK exists in two conformations, with either *cis* or *trans* Pro300, and with intrinsically different activation kinetics for the two species, this can lead to the bi-phasic activation of SthK (Fig. 1f), with *cis* Pro300 being the slow activating, and *trans* Pro300 the fast-activating species. Substituting Pro300 with Ala or Val would eliminate the slow-activating *cis* species, with all channels now adopting only the fast-activating *trans* conformation, as seen above (Fig. 1f). If this model is correct, peptidyl-prolyl *cis/trans* isomerases (prolyl isomerases, PPIases)[24] should accelerate the activation kinetics. PPIases are grouped into three families and participate in processes such as protein folding[22,25], bacterial and viral infections[26,27], gene expression[28,29], and necrosis[30,31] among others.

We assayed the time course of activation of WT SthK in the presence of two different PPIases: SlyD, a bacterial FKBP-type isomerase[32,33] and human, mitochondrial Cyclophilin D (CypD)[34]. Pro300 is preceded by Val299 and using two different PPIases enabled us to avoid potential problems with substrate specificity of these enzymes towards natively folded proteins[35,36]. In the presence of either SlyD or CypD, the slow activation phase of SthK was abolished, and the channels displayed maximal activity after the shortest mixing time, similar to the effect observed when Pro300 was mutated (Fig. 2a, b). The presence of PPIases did not affect the activation time course of SthK P300A (Supplementary Fig. 4b,c), suggesting that Pro300 is the only proline that contributes to this effect and that PPIases do not have non-specific effects on channel activation. Application of purified BSA had little effect on SthK

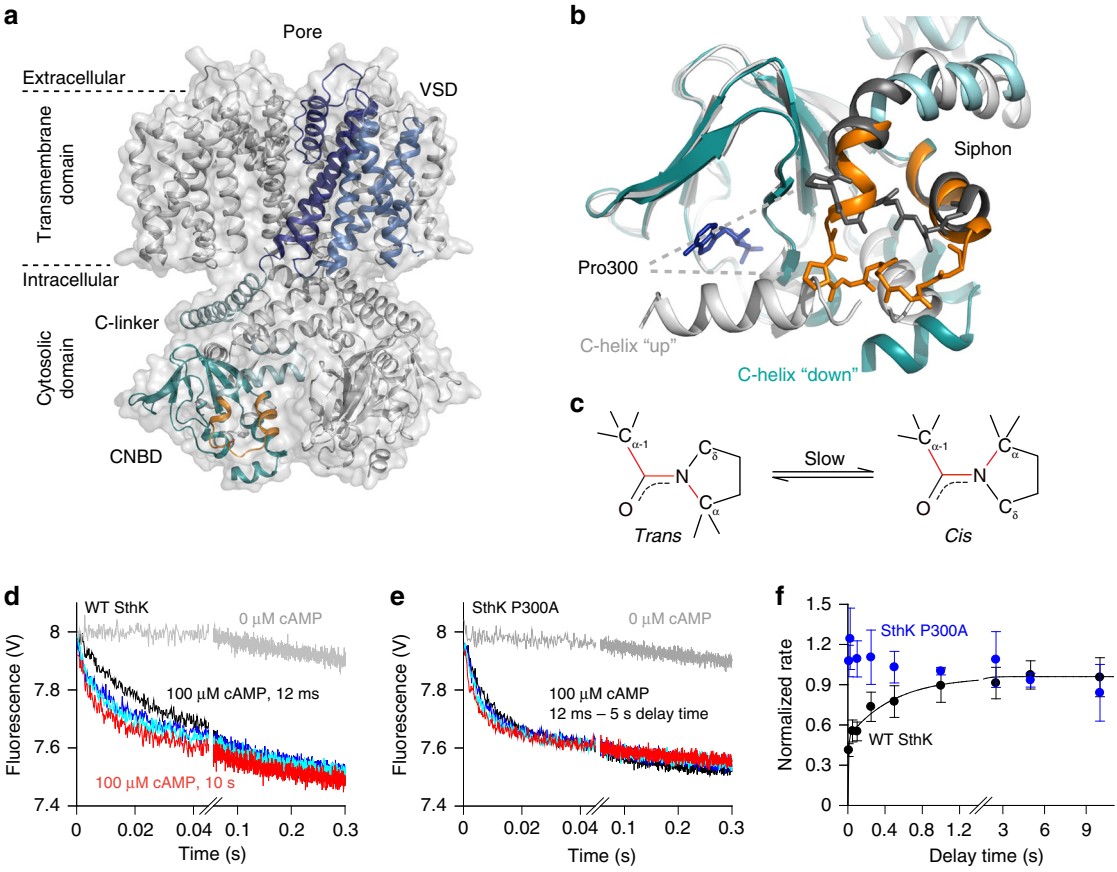

**Fig. 1 Structural and functional overview of SthK. a** SthK ion channel (PDB: 6CJU) in surface (transparent gray) and cartoon representation. The separate domains of one monomer of the protein are shown in different colors with voltage-sensing domain (VSD): blue, pore domain: purple, C-linker: light green, and CNBD: cyan. Siphon is highlighted in orange. **b** Overlay of the CNBD of full-length SthK (PDB: 6CJU) in cyan and the isolated domain (PDB: 4D7T) in gray, with the siphons highlighted in orange and black, respectively. The central loop has residues in stick representation. cAMP is in stick representation (blue) for full-length SthK. **c** *trans* and *cis* conformer of a prolyl bond. **d**, **e** Quenching kinetics from the Tl$^+$ flux assay for WT (**d**) and P300A (**e**) SthK. Samples were incubated with 0 μM cAMP (gray), and with 100 μM cAMP for 12 ms (black), 100 ms (blue), 500 ms (cyan) and 10 s (red, 5 s for SthK P300A). **f** Tl$^+$ flux rates (Eqs. (2) and (3)) obtained from kinetics as shown in (**d**) and (**e**) for WT SthK (black, $n = 8$ independent experiments) and SthK P300A (blue, $n = 5$ independent experiments) as a function of the delay time. Symbols represent mean ± S.D. The activation time course (black curve) was fitted according to a double exponential function with amplitudes $a$ and rate constants $k$ of $a_1 = 0.5 ± 0.04$, $k_1 = 160 ± 50$ s$^{-1}$, $a_2 = 0.45 ± 0.04$, $k_2 = 2 ± 0.5$ s$^{-1}$. Source data are provided as a Source Data file.

activation, ruling out non-specific protein-protein interactions (Supplementary Fig. 4a,c).

To further characterize the effect of PPIases on SthK, we investigated the effect of different isomerase concentrations on the activation kinetics of the channel. As expected for an enzymatic reaction, both PPIases increased the activation rate in a concentration-dependent manner (Fig. 2c, d). This effect reaches saturation at micromolar concentrations of prolyl isomerases. To test if the buffer conditions in the stopped-flow assay affect the catalytic activity of PPIases, we performed a standard isomerization assay on short peptides[36] using the same enzymes and same buffer conditions (Supplementary Fig. 3a, b). The catalytic efficiency of neither PPIase was affected by the conditions used in the stopped-flow assay (Fig. 2f, navy vs gray bars). The assay also confirmed the identity of both isomerases, since they display their characteristic sequence specificity for the residue located prior to the proline (Fig. 2f, filled vs hatched bars)[35,36].

**Cyclosporin A reverses the effect of CypD on SthK.** If the effect of prolyl isomerases on the activation kinetics of SthK is due to their catalytic activity, then specifically inhibiting their catalytic activity should abolish their effect on SthK. Cyclosporin A (CsA) is a cyclic peptide that binds with nanomolar affinity in the

catalytic site of Cyclophilins and inhibits their activity[34,37–39]. Therefore, we tested the effect of CypD on SthK activity in the presence of CsA. SthK activity indeed decreased with increasing concentrations of the inhibitor, with an IC$_{50}$ value of ≥3 μM (Fig. 2e). Importantly, the normalized rate of Tl$^+$ influx approaches the value measured in the absence of PPIase. This indicates that inhibiting the catalytic activity of CypD with CsA reverses the effect on the activation kinetics of SthK.

The concentrations of CsA needed for CypD inhibition in our experiments are higher than the reported nanomolar affinities of CsA for cyclophilins[34,37,38]. This can be attributed to factors such as a relatively high enzyme concentration in our assay, which does not allow for accurate determination of the inhibition constant, or the possibility that CsA inhibits CypD less efficiently under our experimental conditions. We tested whether our experimental conditions are a factor by performing CypD enzyme inhibition assays with CsA using the peptide-based isomerization assay[36] under the same conditions as the stopped-flow assay. We found that while in the absence of liposomes CsA inhibits the activity of CypD with an IC$_{50}$ value of about 13 nM, the IC$_{50}$ value is significantly higher in the presence of liposomes (IC$_{50}$ ≥ 2 μM, Supplementary Fig. 3c). This can be explained by the hydrophobic character of CsA, which leads to interactions with

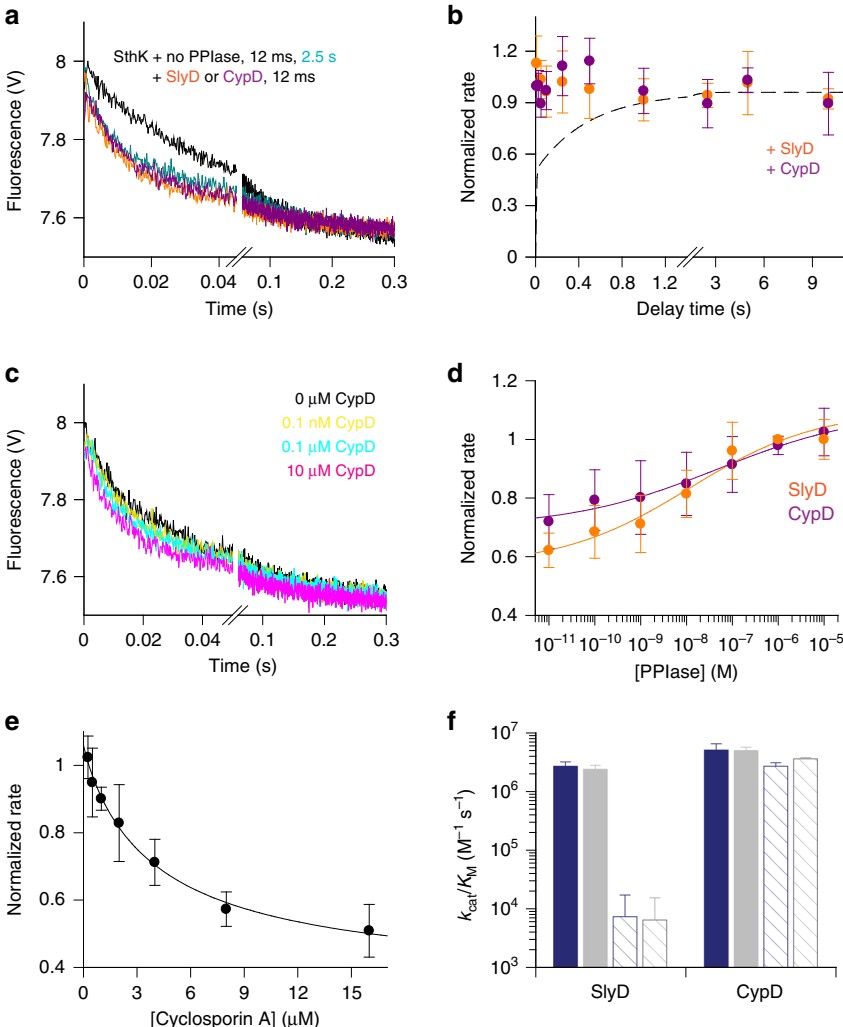

**Fig. 2 Enzymatic activity of PPIases and their effect on SthK. a** Quenching kinetics from the Tl$^+$ flux assay for WT SthK + 100 μM cAMP after incubation for 12 ms (black) and 2.5 s (teal) and after 12 ms in the presence of 1 μM CypD (purple) or 1 μM SlyD (orange). **b** Rate constants (Eqs. (2) and (3)) obtained from kinetics as shown in (**a**) for WT SthK in the presence of 100 μM cAMP and 1 μM CypD (purple, $n$ = 3 independent experiments) or 1 μM SlyD (orange, $n$ = 6 independent experiments). Dashed line represents the activation of SthK in the absence of PPIases, taken from Fig. 1f. **c** Quenching kinetics from the Tl$^+$ flux assay performed in the presence of different concentrations of CypD (0 M CypD—black, 10$^{-10}$ M—yellow, 10$^{-7}$ M—cyan, 10$^{-5}$ M—pink). Kinetics were obtained after 12 ms incubation with 100 μM cAMP. **d** Normalized rate constants (Eqs. (2) and (3)) from kinetics as shown in (**c**) for CypD (purple, $n$ = 4 independent experiments) and SlyD (orange, $n$ = 4 independent experiments). Data were analyzed using Eq. (4) yielding EC$_{50}$ values of ~30 nM. **e** Normalized rate of Tl$^+$ flux for SthK after activation by 100 μM cAMP for 12 ms in the presence of 1 μM CypD and increasing concentrations of CsA ($n$ = 3 independent experiments). Data were fitted to Eq. (5) giving IC$_{50}$ = 4 ± 1.6 μM, $s$ = 0.9 ± 0.2. **f** Catalytic activities $k_{cat}/K_M$ of SlyD and CypD from a peptide-based isomerization assay. Values, plotted as bars, are obtained from fits of data points as shown in Supplementary Fig. 3b for the substrates Abz-ALPF-pNa (filled bars) and Abz-AEPF-pNa (hatched bars). The assays were performed in 20 mM Hepes, 100 mM KCl, pH 7.4 (gray) and 10 mM Hepes, 140 mM KNO$_3$, pH 7.4 (navy). Numerical values together with the S.E. of fits are also in Supplementary Table 1. Symbols in (**b**, **d**, **e**) are mean ± S.D., in (**f**) fitted values ± S.E. of fits. Source data are provided as a Source Data file.

liposomes[40,41], thus decreasing the effective concentration of CsA available for enzyme inhibition.

**Single-channel gating is unaffected by prolyl isomerases.** PPIases have been reported to interact with different ion channel proteins, to regulate the function of ion channels and to act as scaffolding proteins between channels and other interaction partners[42–46]. For example, PPIases have been shown to modulate the conductance and open probability of certain ion channels[45,47]. To test if the interaction of PPIases with SthK also affects intrinsic properties of SthK such as single-channel conductance and the open/closed equilibrium, we performed single-channel recordings in the absence and presence of CypD at saturating concentrations of cAMP, to restrict the gating to the fully-liganded open and closed states (Fig. 3a). Under these conditions, 1 μM CypD, a concentration sufficient to increase the channel activation rate (Fig. 2d), did not change the single-channel properties of SthK (Fig. 3a, b, Supplementary Fig. 3d, e). Furthermore, the presence of PPIase does not lead to spontaneous channel openings in the absence of cAMP (Fig. 3a). This indicates that under steady-state conditions, isomerases do not affect the SthK single-channel characteristics or the intrinsic open/closed gating equilibrium.

**The affinity of SthK for cAMP changes over time.** In the presence of PPIases, WT SthK shows only fast activation (Fig. 2b), similar to SthK P300A, an all-*trans* mimic (Fig. 1f). In the context of a model where the channel exists as a mixture of two species

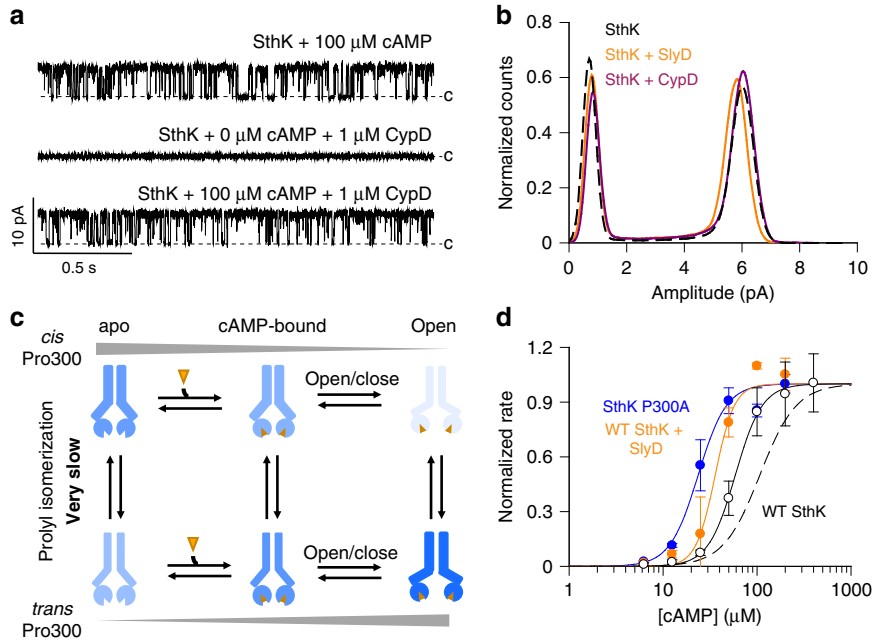

**Fig. 3 Effect of PPIases on the gating of SthK. a** Representative single-channel recordings of WT SthK at +100 mV in the presence of 100 µM cAMP, 0 µM cAMP + 1 µM CypD, and 100 µM cAMP + 1 µM CypD, as indicated. Dashed lines indicate the closed level. **b** Normalized all amplitude histogram from single-channel recordings of WT SthK in the presence of 100 µM cAMP, without PPIase (black, dashed line) and in the presence of 1 µM SlyD (orange) or 1 µM CypD (purple) at +100 mV. These experiments were performed three times with similar outcome. **c** Schematic state model (blue cartoon is the channel, orange triangle is the ligand) for the proline switch in SthK. Column labels indicate apo, cAMP-bound, and open states. Row labels indicate *cis* and *trans* Pro channel forms. Vertical transitions are modulated by prolyl isomerization. Gradient lines indicate the shift in the *cis/trans* equilibrium between the apo and the open state. **d** Initial rates of Tl[+] flux obtained from the stopped-flow assay (Eqs. (2) and (3)) are plotted as functions of the cAMP concentration. Lines represent fits according to Eq. (4). Data with 2.5 s delay time are shown for SthK P300A in blue ($EC_{50}$ = 24 ± 2 µM, $n_H$ = 2.9 ± 0.6, $n$ = 3 independent experiments), WT SthK in the presence of 1 µM SlyD in orange ($EC_{50}$ = 36 ± 2 µM, $n_H$ = 4 ± 0.9, $n$ = 3 independent experiments). The fit for WT SthK in the absence of PPIase is shown as dashed line. WT SthK after about 45 min delay time is shown as open circles and the fit as continuous black line ($EC_{50}$ = 59 ± 1 µM, $n_H$ = 3 ± 0.9, $n$ = 4 independent experiments). Symbols represent mean ± S.D. Source data are provided as a Source Data file.

with *cis* or *trans* Pro300 that can interconvert, this can be explained by assuming that the *trans* Pro species has higher affinity for cAMP. Thus, upon ligand binding, the *cis* species will be driven towards the open state with *trans* Pro300 (model in Fig. 3c). Indeed, the apparent affinity for cAMP in SthK P300A (all-*trans* mimic) is higher than for WT SthK in the stopped-flow assay ($EC_{50}$ for SthK P300A is 24 µM, compared to $EC_{50}$ for WT SthK of 107 µM (Fig. 3d, Supplementary Fig. 4d and[11]). It is important to note that in these experiments, channels were activated with cAMP for only 2.5 s before the activity was measured. Because prolyl isomerization is an intrinsically slow reaction[22], we increased the incubation time for WT SthK with cAMP to approximately 45 min, to allow for a potential *cis/trans* re-equilibration (vertical transitions in Fig. 3c). The $EC_{50}$ value is decreased from 107 µM to ~59 µM after 45 min of activation, but it is still higher than for SthK P300A after 2.5 s (Fig. 3d). To test if the *cis/trans* re-equilibration at Pro300 is complete after 45 min, we next measured the dose-response curve in the presence of prolyl isomerases, which should decrease the kinetic barrier of prolyl isomerization. After 2.5 s activation time in the presence of 1 µM SlyD, the $EC_{50}$ value of WT SthK for cAMP is shifted to 35 µM, close to the $EC_{50}$ for SthK P300A (24 µM). These results show that the *cis/trans* equilibrium at Pro300 is indeed shifted during the activation of SthK and that *trans* Pro300 is favored in the active state (Fig. 3c, d, Supplementary Fig. 4d, e). It is important to note that we have no information about the number of subunits within a channel tetramer that need to have *cis* Pro300 in order to elicit the reported effects of slow activation and low ligand-binding affinity.

**Structural characterization of SthK P300A**. Our data predict that WT SthK exists as a mixture of channels with Pro300 in either *cis* or *trans*. Accordingly, structural analyses using cryoEM should ideally detect different channel conformations that differ in the configuration of Pro300. However, even if the resolution was high enough (~2.2 Å) in the siphon region to assign the backbone angles at the proline, classification and alignment algorithms would have extreme difficulty separating between subunits that only differ in the conformation of one peptide bond. Our previously solved structures of WT SthK[13] display 3.3–3.5 Å overall resolution, with ~4 Å local resolution in the siphon region (Supplementary Fig. 8e), indicating flexibility in this loop. Predictably, only one closed-state channel conformation was detected[13], which we hypothesize is an average over molecules with *cis* and *trans* Pro300. The structural heterogeneity at Pro300 likely contributes to the flexibility and low local resolution in this part of the channel.

Here, to overcome this heterogeneity and understand what the structure could look like with only one, defined configuration at Pro300, we solved the single-particle cryoEM structure of SthK P300A (all-*trans* Pro mimic) in lipid nanodiscs and in the presence of cAMP (Fig. 4 and Supplementary Fig. 5). In agreement with the low open probability of SthK P300A at 0 mV (Supplementary Fig. 2c), we observed that the majority of SthK P300A particles (~90%), adopt a closed-state conformation (3.4 Å resolution), similar to cAMP-bound WT SthK[13]. About 9% of the particles for SthK P300A adopt a different conformation, which we call a putatively open state because it shows structural changes consistent with an open channel but the resolution is too

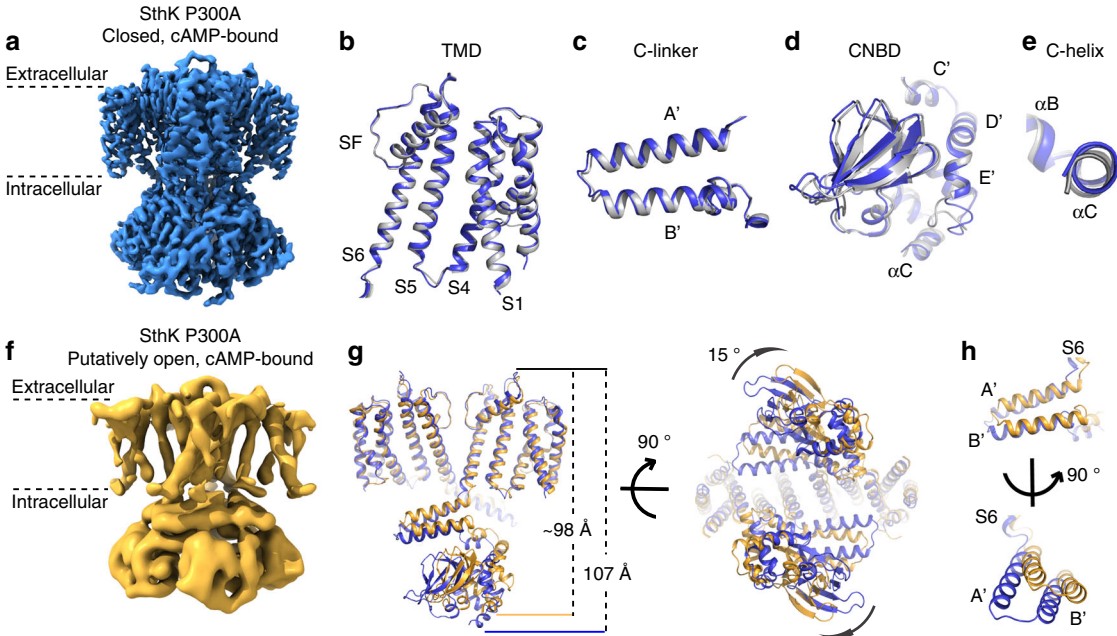

**Fig. 4 Cryo-EM structures of SthK P300A. a** Density map of SthK P300A bound to cAMP in the closed state. **b–e** Comparison of WT SthK (PDB: 6CJU, gray) and SthK P300A (blue) in the closed state. Both structures were aligned to the turrets of WT SthK. Comparison of the TM domains is shown in (**b**), helices A′ and B′ of the C-linker are in (**c**), the CNBDs in (**d**), and the C-helix in (**e**). **f** Density map of SthK P300A in the putatively active state. **g, h** Comparison between SthK P300A in the closed state (blue) and the putatively active state (yellow) with both structures aligned to the turrets of the closed state. **g** Side and bottom view of two subunits. **h** Helices A′ and B′ of the C-linker.

low to determine the diameter of the intracellular entrance (6.7 Å resolution, Fig. 4 and Supplementary Figs. 5–8). For WT SthK an open state was also expected but not detected, possibly due to minor differences in the sample preparation or a more dynamic open conformation for WT SthK, added to the difficulties of identifying a low-populated state from a small number of particles with existing algorithms.

Analysis of the major, closed conformation of SthK P300A, revealed that the siphon, where the mutation is located, has a slightly different conformation compared to WT SthK (Supplementary Fig. 8e). However, the density in this area is weak, making a detailed comparison difficult. Interestingly though, the density in the siphon region of SthK P300A appears similar in resolution to the density around it, indicating less heterogeneity than in the WT channel, as one may expect if *cis/trans* Pro300 was contributing to the flexibility (Supplementary Fig. 8e). Overall, the closed states of SthK P300A and WT SthK[13] overlay with an RMSD of ~0.7 Å indicating that the structures are similar, but not identical (Fig. 4a–e, Supplementary Fig. 8a–c). When they are aligned only on the TM domains, the TM domains superimpose with an RMSD of ~0.4 Å (Fig. 4b) meaning that they are now identical. In this alignment, the cytosolic C-linker/CNBD domains in SthK P300A are now globally displaced by a small amount from the WT SthK structure (RMSD ~ 1.5 Å). This displacement is a rigid body movement, that brings the CNBDs in SthK P300A about 1–1.5 Å closer to the membrane compared to their position in WT SthK (Fig. 4c–e). The structure of SthK P300A in the closed state reveals a previously undetected conformation of SthK, which we hypothesize is what WT SthK with all-*trans* Pro300 would look like.

The minor channel conformation identified in the SthK P300A dataset, although much lower resolution (6.7 Å), shows large conformational changes when compared to the closed-state structure (Fig. 4f–h), using again the TM domains for the alignment. The structure is shorter and wider (Fig. 4f, g), with an increased diameter at the intracellular entryway reminiscent of an open channel (Fig. 4g and Supplementary Fig. 6i). The C-linker is rotated clockwise and shifted outwardly (Fig. 4h)[15] by ~15° relative to the channel pore, previously observed in an imaging study[12], and leads to a displacement of the CNBDs. The C-helix is shifted upwards, with a simultaneous upwards movement of the siphon by 5 Å. An overlay of the CNBD in this state with the X-ray structure of the isolated CNBD in the presence of cAMP[48] suggests that the CNBDs are in an activated state (Supplementary Fig. 8d). Although the resolution of this conformation is low, the conformational changes in the C-linker/CNBD domains and the increase in radius at the intracellular gate are structural features of an open-activated cyclic nucleotide-modulated channel[18,49] and raise the possibility that this minor conformation represents an open state.

## Discussion

Prolyl isomerization is mostly known as a rate-limiting step during protein folding reactions, but it also works as a tool to fine-tune activity in natively folded proteins[22,50,51]. Only a few reports in the literature suggest prolyl isomerization as a mechanism to modulate ion channels, with little to no follow-up studies (for 5-HT$_3$ receptors and TRPC1 channels[45,52,53]). PPIases have also been proposed to regulate the function of ion channels by forming complexes[42,43,45,54–58] and to be important for their functional expression[46]. Most prominently, ryanodine receptors (RyR) were shown to interact tightly with FKBPs[54,59], but it was not investigated whether their enzymatic activity, the isomerization of prolyl bonds, was actually involved[47].

Here, we show that a conserved proline residue in the ligand-binding domain of SthK, a cyclic nucleotide-gated channel, undergoes *cis/trans* isomerization upon activation of the channel with cAMP. Unlike previous reports, we used purified proteins for both the enzymes and the substrate (prolyl isomerases and SthK channels), functional and structural assays and enzymatic catalysis studies, allowing us to construct a detailed mechanistic

picture of how prolyl isomerization modulates the activation kinetics of SthK channels.

We employed kinetic models to understand mechanistically how activation of SthK is affected by prolyl isomerization. The simplest gating mechanism for ligand-gated ion channels requires at least 3 states: an apo, a ligand-bound, and an active state (Fig. 3c, Supplementary Fig. 9a). However, to explain the bi-phasic activation of SthK channels with cAMP, two channel forms are needed, which activate with different kinetics upon cAMP application (with Pro300 in either *cis* or *trans* configuration, Fig. 3c). The slow phase is abolished in the presence of PPIases (Fig. 2b) which means that the two channel species can interconvert (vertical transitions in model in Fig. 3c). The *cis/trans* re-equilibration upon cAMP application is slow (Fig. 3d), and after short activation times (2.5 s) the EC$_{50}$ value for WT SthK thus reflects the weighted sum of contributions from the two independent protein species (Eq. (1)).

$$X \cdot EC_{50}^{trans} + (1 - X) \cdot EC_{50}^{cis} = EC_{50}^{apparent} \qquad (1)$$

with X being the fraction of *trans* Pro300 and EC$_{50}^{trans}$, EC$_{50}^{cis}$, EC$_{50}^{apparent}$ the apparent affinities for cAMP for the *trans* and the *cis* species, and the overall measured apparent affinity. Since SthK P300A, an all-*trans* mimic, activates with fast kinetics and requires lower concentrations of cAMP (Fig. 3d), the *trans* Pro form has fast activation and higher ligand-binding affinity. From the amplitudes of the bi-phasic activation we estimate that in apo SthK the fast-activating species (with *trans* Pro300) is populated to about 40%[11]. Using Eq. (1) and EC$_{50}^{P300A}$ = EC$_{50}^{trans}$ = 24 μM we estimate the apparent affinity for the *cis* form to be: EC$_{50}^{cis}$ ~ 160 μM.

We tested two different models. Since our experiments do not provide information about microscopic rate constants, we did not aim to reproduce the exact activation rates and EC$_{50}$ values obtained experimentally, but rather aimed to qualitatively capture the key trends, such as the changes in activation kinetics and EC$_{50}$ values. The first model (Supplementary Fig. 9b) posits that the *cis* Pro300 channel form rarely opens, and most channel activity comes from the *trans* Pro300 channel form. Upon application of cAMP, the *trans* Pro300 channels activate fast, but maximum channel activity is reached only after all molecules with *cis* Pro300 have isomerized to *trans*. In this model prolyl isomerization is an on/off switch between *cis* and *trans* Pro300 SthK forms. This model can explain the bi-phasic activation but cannot account for the slow shift in the EC$_{50}$ values since the population of the active channel species is directly linked to prolyl isomerization (Fig. 3d and Supplementary Fig. 9c–e). In addition, the rate for uncatalyzed prolyl isomerization in this model has to be unusually fast to be complete within ~2 s, however, such isomerization processes generally happen within tens to hundreds of seconds[22].

In the second model, we assume that prolyl isomerization is modulatory rather than an on/off switch, meaning that it allows switching between two-channel forms that both display activity (Fig. 3c and Fig. 5a). Here, both forms can be activated by cAMP, albeit with different affinities and different kinetics for the *cis* and *trans* Pro300 channel species (compare Fig. 5b with Figs. 1f and 2b). At low concentrations of cAMP, the high-affinity *trans* Pro300 form will be predominantly activated and the shift of the *cis/trans* equilibrium occurs in the apo state (vertical transition between states 1 and 2, Fig. 5a), since state 2 is depleted during channel activation. With increasing cAMP concentration, the lower-affinity *cis* Pro300 species will also activate and the *cis/trans* equilibrium shift occurs also in the open state (vertical transition between states 5 and 6, Fig. 5a), after maximum channel activity has been reached (Fig. 5b, Supplementary Fig. 9f). Vertical transitions between states 3 and 4 were omitted in order to reduce the number of parameters as their inclusion (with the same

kinetic rates as for the transitions between states 5 and 6) did not change the final outcome. Simulations using the modulatory mechanism capture not only the bi-phasic activation but also the time-dependent shift of the EC$_{50}$ values (Fig. 5c and Supplementary Fig. 9f–i). In this model, *cis* and *trans* Pro300 SthK display similar open-closed equilibria (transitions 3–5 and 4–6), and maximum channel activity depends on the affinities of the two species, rather than the conformational state of Pro300 (as in the on/off switch model). However, *trans* Pro300 SthK is still the high-affinity species and the *cis/trans* equilibrium is thus slowly shifted towards this form, which happens much more slowly than either the *cis* or *trans* channels activate, giving rise to the time-dependent shift in EC$_{50}$. We propose a mechanism where *cis/trans* heterogeneity at Pro300 leads to different affinities for ligands and in turn to different activation kinetics of the two forms of SthK that can be fine-tuned by prolyl isomerases. This mechanism allows adaptation of the electrical response elicited by SthK to a wide range of physiological conditions.

In our experiments, *cis/trans* re-equilibration is slow and not complete even after 45 min (Fig. 3d) suggesting that this process is unlikely to happen on its own under physiological conditions. However, in the presence of PPIases the kinetic barrier imposed by prolyl isomerization is reduced and the EC$_{50}$ value after short delay times is close to the EC$_{50}$ value of SthK P300A (Fig. 3d). This EC$_{50}$ value is a measure for the *cis/trans* equilibrium in the active state. From Eq. (1) we obtain a *trans* Pro300 content of roughly 0.9 in the open state (EC$_{50}^{trans}$ = 24 μM, EC$_{50}^{cis}$ = 160 μM and EC$_{50}^{apparent}$ = 35 μM). The *cis/trans* equilibrium at Pro300 in SthK is thus shifted from about 60/40 in the apo state to 10/90 in the open state (equivalent to ΔΔG$^{cis/trans}$ ~ 6.5 kJ/mol).

The slow activation phase of SthK with cAMP, which we attribute to a sizable amount of *cis* Pro300 channels, was not observed by Morgan et al.[14]. However, the experiment they performed did not employ purified SthK protein under defined conditions, but rather involved patch-clamp recordings of *E. coli* spheroplasts. Cellular cAMP and PPIases could easily precondition SthK by shifting the *cis/trans* equilibrium towards the fast-activating *trans* Pro300 species and slow activation could no longer be observed (Figs. 2b, 3d).

*Cis* Pro300 can be modeled into the cryoEM density of WT SthK indicating that this conformation is consistent with the experimental density (Fig. 5d). In the crystal structure of the isolated C-linker/CNBD domain, Pro300 is either *trans* or not resolved in the presence of cAMP and cGMP[48], consistent with our findings that Pro300 adopts predominantly the *trans* conformation in the active state. Inversely, in apo SthK the *cis/trans* equilibrium at Pro300 is shifted towards the intrinsically less favored *cis* conformation and additional interactions need to be established to provide enough energy for this shift. We estimate this energy to be ΔΔG$^{cis/trans}$ ~ 6.5 kJ/mol which is similar to the formation of two hydrogen bonds[60]. Close inspection of the interface between adjacent CNBDs reveals subtle rearrangements in the closed-state of SthK P300A compared to WT SthK, raising the possibility that this energy could originate from different CNBD-CNBD interactions with *cis* or *trans* Pro300 (Supplementary Fig. 8f). This could indicate that the oligomerization state of the CNBDs might also influence the *cis/trans* equilibrium at Pro300[61]. Pro300 is centrally located to sense different sets of interactions (Fig. 5e), which can translate into different activation kinetics.

Here, we present prolyl isomerization as a mechanism to regulate the function of a cyclic nucleotide-modulated ion channel by enabling the existence of two different forms of apo SthK: 1) with high affinity for cAMP and fast activating (*trans* Pro300), and 2) with low cAMP binding affinity and slow

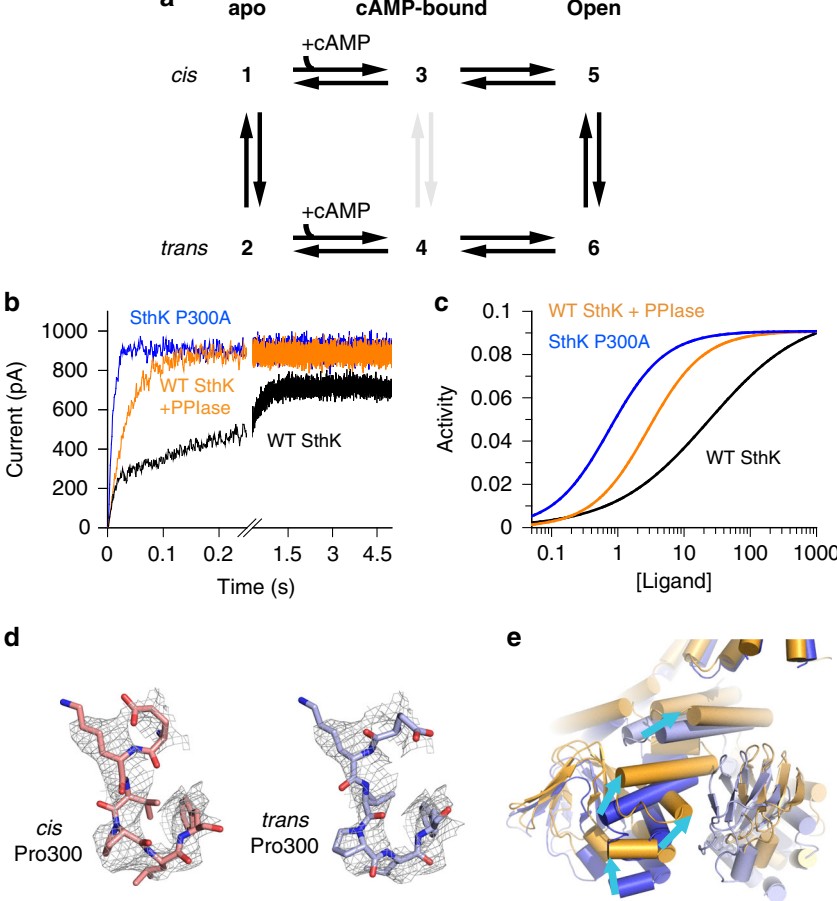

**Fig. 5 Mechanistic and structural implications of the proline switch in SthK. a** Model of the modulatory prolyl isomerization in SthK used for simulations as shown in (**b**, **c**), and Supplementary Fig. 9f–i. Transitions in gray were omitted for the simulations since their inclusion does not change the outcome and to reduce the number of parameters. **b** Simulated, macroscopic activation time courses of 10,000 channels upon ligand application (100 μM cAMP) according to the model shown in (**a**). **c** Comparison of the theoretical dose-response curves of WT SthK (black), WT SthK + PPIase (orange), and SthK P300A (blue) after 2.5 s activation by cAMP (same condition as the experimental stopped-flow assay shown in Fig. 3d). Conditions and results of simulations are given in Supplementary Table 3 and 4. **d** Modeled *cis* Pro300 (left panel) in comparison to *trans* Pro300 (right panel) in WT SthK (PDB: 6CJU, EMDB: 7484). **e** Propagation of conformational changes during gating of SthK through one CNBD and along the subunit interface. The closed state of SthK P300A is shown in blue, the putatively active state in yellow.

activating (*cis* Pro300). The two species are about evenly populated, which ensures that at low ligand concentrations sufficient channels are being activated to elicit electrical signaling. On the other hand, when the channels are exposed to high ligand concentrations, both species are activated; however, the maximum response is established slowly. As the apo state with *trans* Pro300 is depleted during activation and the active states of both *trans* and *cis* species are populated, the *cis/trans* equilibrium is slowly shifted towards *trans* due to its higher binding affinity. This shift is unlikely to happen on its own on a physiological time scale, but it can be efficiently catalyzed by prolyl isomerases. The presence of PPIases provides an additional level of regulation, which allows for shifting the *cis/trans* equilibrium and thus altering the electrical output within milliseconds. Taken together, we suggest that native-state prolyl isomerization acts like a built-in molecular pacemaker, which, in cells, can be modulated by prolyl isomerases. Conservation of this proline among SthK and HCN channels, which display slow current activation with cAMP, but not in CNG channels, which display fast current activation with cyclic nucleotides, raises the intriguing possibility that this mechanism may also be at work in HCN channels.

## Methods

**Protein expression and purification**. SthK (UniProt G0GA88) was expressed as described in ref. [11]. Briefly, protein expression was performed in *E. coli* C41 (DE3) cells (Lucigen) at 20 °C, overnight. Cells were broken by sonication and membranes were solubilized by 30 mM *n*-dodecyl-β-d-maltopyranoside (DDM, Anatrace). SthK was purified by immobilized metal affinity chromatography using a 5 ml HiTrap column (GE Lifesciences) charged with $Co^{2+}$. The protein was concentrated to ~10 mg/ml (Amicon® Ultra-15, Millipore) and applied to gel filtration (Superdex200 10/300, GE Lifesciences) in 20 mM Hepes, 140 mM $KNO_3$, 0.5 mM DDM, pH 7.4 (Supplementary Fig. 1d,e). The purified protein was used immediately for reconstitutions into large unilamellar vesicles or flash frozen and stored at −80 °C for future use. The entire purification was performed at 4 °C. Mutations in the SthK gene for P300A and P300V were introduced by Quikchange PCR using Q5 Polymerase (NEB) and the primers listed in Supplementary Table 5. Protein expression and purification of both variants were performed as for WT SthK.

The gene for mitochondrial Cyclophilin D (UniProt P30405, residues 30-207, in the following called CypD) was cloned into pET11a using NdeI and BamHI restriction sites carrying an initial Met and a C-terminal GGSGSG-His$_6$ purification tag leading to pET11a-CypD (Supplementary Table 5). SlyD (UniProt P0A9K9, residues 1–165, in the following called SlyD) was expressed from pET24a as a C-terminal His$_6$ fusion[33].

*E. coli* BL21 (DE3) cells (NEB) were transformed with either pET24a-SlyD or pET11a-CypD and grown in 2 L of LB media in the presence of 50 μg/ml Kanamycin or 100 μg/ml Ampicillin, respectively, at 37 °C (220 rpm) to an $OD_{600}$ of 0.6. A final concentration of 1 mM IPTG was used to induce protein expression for 4 h before the cells were collected by centrifugation (4000 *g*, 4 °C, 10 min), resuspended in breaking buffer (50 mM Tris, 100 mM KCl, pH 8) and broken by

sonication in the presence of 1 mg DNaseI (Sigma-Aldrich), 1 mg lysozyme (Sigma-Aldrich), 85 μg/ml PMSF (Roche) and cOmplete ULTRA mini protease inhibitor (Roche). The suspension was cleared by centrifugation (36,000 $g$, 4 °C, 45 min), the supernatant was passed through a 0.22 μm filter and applied to a 5 ml HiTrap column (GE Lifesciences) charged with Ni$^{2+}$ equilibrated in 20 mM Hepes, 100 mM KCl, pH 7.8. Nonspecifically bound proteins were removed by washing with 50 ml of wash buffer (20 mM Hepes, 100 mM KCl, 50 mM Imidazole, pH 7.8) and the protein was eluted with elution buffer (20 mM Hepes, 100 mM KCl, 250 mM Imidazole, pH 7.8). The eluate was concentrated to 1.2 ml using a 3.5 kDa cutoff (Amicon® Ultra-15, Millipore).

Both enzymes were further purified by gel filtration (Superdex200 16/600, GE Lifesciences) in 20 mM Hepes, 100 mM KCl, pH 7.4. Protein containing fractions were pooled and concentrated to 2 ml. The final protein concentration was determined photometrically using extinction coefficients $\varepsilon_{280}$ of 5960 M$^{-1}$ cm$^{-1}$ for SlyD and 9970 M$^{-1}$ cm$^{-1}$ for CypD.

Bovine serum albumin (BSA, Sigma Aldrich) was used in control experiments. We detected some impurities in commercially available BSA. To further purify BSA, we subjected resuspended protein to gel filtration (Superdex200 16/600, GE Lifesciences) in 20 mM Hepes, 140 mM KNO$_3$, pH 7.4 before using it in our assays.

**Stopped-flow Tl+ flux assay.** All stopped-flow experiments were performed at 25 °C using a SX.20 sequential mixing fluorescence spectrophotometer (Applied Photophysics, Leatherhead, UK) controlled using ProDataSX as described before[62]. In order to prepare liposomes for the Tl$^+$ flux assays 15 mg of DOPC:POPG: Cardiolipin (5:3:2, w/w/w) in chloroform were dried to a thin film in glass tubes under constant N$_2$ flow and further dried under vacuum overnight. The next day lipids were rehydrated in reconstitution buffer (15 mM Hepes, 150 mM KNO$_3$, pH 7.4) to reach a concentration of 13.46 mg/ml and solubilized by the addition of CHAPS (33 mM final concentration) while sonicating the solution in a water bath. ANTS (8-aminonaphthalene-1,3,6-trisulfonic acid, disodium salt, Life Technologies, 75 mM in ddH$_2$O, pH 7.4) was added to a final concentration of 25 mM in a total of 3500 μl. Purified protein after gel filtration was added (30 μg/mg lipid) and incubated for 20 min. Detergent removal was performed by adding 0.7 g SM-2 BioBeads (BioRad) pre-equilibrated in 10 mM Hepes, 140 mM KNO$_3$, pH 7.4. The sample was incubated at 21 °C under constant agitation for 3 h, the supernatant was transferred to a new glass tube and stored at 13 °C overnight. In order to remove extravesicular ANTS the liposome solution was briefly sonicated in a water bath sonicator, extruded through a 0.1 μm filter (Whatman) using a mini-extruder (Avanti Polar lipids) and then passed over a PD-10 desalting column (GE Lifesciences) equilibrated in sample buffer (10 mM Hepes, 140 mM KNO$_3$, pH 7.4). The liposome solution was diluted 5-times in sample buffer right before the experiments to ensure a good signal-to-noise ratio. The entire reconstitution was performed in the absence of cAMP to silence channels that incorporate with their CNBDs in the lumen of the vesicles. Accordingly, only channels that face the outside of the vesicles with their cytosolic domains can be activated by externally applied cAMP (cAMP is not membrane permeable, logP = −3.4 as calculated according to ref. [63]) and are substrates for PPIases.

For the activation time course SthK incorporated in liposomes was mixed 1:1 with sample buffer containing 200 μM cAMP (in order to reach 100 μM cAMP after the first mixing event) and incubated for the desired time (12 ms–10 s) before the second mixing was performed with quenching buffer (10 mM Hepes, 90 mM KNO$_3$, 50 mM TlNO$_3$, 100 μM cAMP, pH 7.4) (Supplementary Fig. 1b, c). For the time course of activation in the presence of PPIases 2 μM of enzyme were added to the liposome solution and incubated for at least 10 min before the experiment yielding 1 μM of enzyme in the aging loop (after the first mixing event). The quenching buffer did not contain PPIase.

The dose-response curves for cAMP, PPIase, or inhibitor were determined in a similar assay. SthK incorporated in liposomes was mixed (1:1) with pre-mixing buffer supplemented with increasing cAMP concentrations and incubated for 2.5 s. The second mixing step (1:1) was performed with quenching buffer containing the final concentration of cAMP. For the 45 min delay experiment, liposomes were pre-incubated with the desired cAMP concentrations. During the two mixing steps the cAMP concentration was kept constant and the delay time was set to 1 s. PPIase concentrations were varied by serial dilution and SthK containing liposomes were incubated with the desired PPIase concentration for at least 10 min before the experiment. The concentration of inhibitor was also varied by serial dilution as 1000-fold concentrated solution in DMSO. 2 μl of these pre-dilutions were diluted in 2 ml of SthK containing liposomes in order to avoid distortion of the liposomes by DMSO. The highest Cyclosporin A (CsA) concentration used in these experiments was 16 μM CsA due to the limited solubility of CsA in aqueous buffer. For the concentration dependence of PPIase and CsA the delay time was set to 12 ms as the effect of PPIase activity on SthK activation is most pronounced after shorter incubation times with cAMP. All presented quenching kinetics are averaged over at least five technical repeats of a representative experiment.

**Analysis of stopped-flow Tl+ flux kinetics.** During one experiment, at least seven technical repeats were performed for each data point, every repeat was visually inspected for its quality, mixing artifacts were sorted out and the remaining repeats (typically 6–7 technical repeats) were analyzed separately in Matlab using a

stretched exponential function (Eq. (2)) and the rate of Tl$^+$ influx into liposomes was calculated at 2 ms according to Eq. (3). 5000 data points were recorded over 1 s, however, to obtain the initial rate of Tl$^+$ influx only the first 100 ms were analyzed. The use of a stretched exponential function is necessary since the sizes of the liposomes vary and different amounts of channel proteins can be incorporated in different liposomes.

$$F_t = F_\infty + (F_0 - F_\infty) \cdot e^{\left\{ -\left(\frac{t}{\tau}\right)^\beta \right\}} \tag{2}$$

$$k_t = \left(\frac{\beta}{\tau}\right) \cdot \left(\frac{0.002\text{s}}{\tau}\right)^{(\beta-1)} \tag{3}$$

with $F_t$, $F_\infty$, $F_0$ being the fluorescence at time t, the final fluorescence and the initial fluorescence, respectively. t is the time (in s), $\tau$ the time constant (in s) and $\beta$ the stretched exponential factor. $k_t$ is the calculated rate (in s$^{-1}$) of Tl$^+$ influx at 2 ms.

The calculated rates $k_t$ were then averaged and the standard deviation (S.D.) was determined. Each experiment was independently repeated at least three times, the obtained rate constants were normalized, averaged and the S.D. was determined. The exact number of independent repeats for each experiment ($n$) is given in the figure legends.

All further analysis was performed in GraFit. For the time course of channel activation, the rates were plotted as a function of the delay time in the first mixing step and fitted according to a double exponential. For the dose-response curve of channel activation by cAMP (EC$_{50}$) the Tl$^+$ flux rates were plotted as a function of the cAMP concentration and analyzed using a modified Hill equation (Eq. (4)),

$$y([\text{cAMP}]) = \frac{\frac{k}{k_{max}} \cdot [\text{cAMP}]^{n_H}}{(\text{EC}_{50})^{n_H} + [\text{cAMP}]^{n_H}} \tag{4}$$

with y being the normalized rates as function of the cAMP concentration (μM), $k$ the rate at a given cAMP concentration, $k_{max}$ the maximum rate, $n_H$ the Hill parameter and EC$_{50}$ the cAMP concentration of half-activation in μM.

The rates obtained from the PPIase inhibition experiments were averaged and plotted as function of the inhibitor concentration. The IC$_{50}$ value was determined by fitting a four parameter logistic equation (Eq. (5)) to the data.

$$y([\text{CsA}]) = \frac{k_{max} - k_0}{1 + \left(\frac{[\text{CsA}]}{\text{IC}_{50}}\right)^s} \tag{5}$$

with y being the Tl$^+$ influx rate as function of the CsA concentration, $k_{max}$ the maximum rate (without inhibitor) and $k_0$ the rate with saturating inhibitor. [CsA] is the concentration of CsA in μM, IC$_{50}$ the inhibition constant in μM and s the slope factor.

**Bilayer recordings.** Single-channel recordings of SthK were performed in a horizontal lipid bilayer setup at room temperature. The cis (upper) and trans (lower) chambers are separated by a partition with a 100 μM diameter hole. 1,2-diphytanoyl-sn-glycero-3-phosphocholine (DPhPC, Avanti Polar Lipids) in chloroform was dried under constant N$_2$ flow, washed in n-pentane, dried again and re-solvated in n-decane to reach a final concentration of 8 mg/ml. Bilayers were formed by using a small air bubble on a 10 μl pipet tip and gently painting over the hole. Proteo-liposomes were thawed, briefly sonicated, and applied to the cis side of the chamber while applying +100 mV current in order to monitor channel incorporations into the bilayer. All bilayer recordings were performed in 10 mM Hepes, 100 mM KCl, pH 7.4. Only the trans chamber buffer was supplemented with cAMP in order to silence channels incorporated with the cytosolic domains facing the cis side. For recordings in the presence of PPIase the trans chamber was additionally supplemented with enzyme. All electrophysiological data were recorded with an Axopatch 200B (Molecular Devices), filtered online at 2 kHz with an eight-pole, low-pass Bessel filter and digitized at 20 kHz (Digidata 1440 A, Molecular Devices). Recordings were controlled using Clampex 10.7.0.3 (Molecular Devices) and analyzed in Clampfit 10.7.0.3 (Molecular Devices) with no additional filtering.

**Peptide-based isomerization assay.** Activity of prolyl isomerases was tested using a FRET based peptide assay[36]. Abz-Ala-Xaa-Pro-Phe-pNA (Abz = 2-aminobenzoyl, Xaa = any amino acid, pNA = para-nitroaniline, here we used Glu and Leu in front of the proline) was dissolved in 0.55 M LiCl/TFE (anhydrous trifluoro-ethanol, Sigma) to reach a 750 μM stock solution of peptides. Under these conditions, the cis proline content is increased and the quencher (pNA) is in close proximity to the fluorophore (Abz). Upon a solvent jump into aqueous buffer (20 mM Hepes, 100 mM KCl, pH 7.4 or 10 mM Hepes, 140 mM KNO$_3$, pH 7.4) the cis/trans equilibrium at proline is shifted towards trans and the pNA moiety moves away from Abz. Accordingly, the fluorescence increase reflects the cis/trans re-equilibration kinetic of the Xaa-Pro bond[36]. The assay was performed at room temperature using a Horiba PTI QuantaMaster™ fluorescence spectrophotometer (excitation: 316 nm, emission 416 nm, 5 mm band width each) controlled using FelixGX.

To determine the activity of PPIases, peptides with either Leu or Glu preceding the Pro were used. Buffer and the desired concentration of PPIase were mixed in a QS quartz cuvette (Hellma Analytics) and incubated for 2 min under constant

stirring. Peptide was added and the fluorescence increase was monitored. Kinetics were fitted with a mono-exponential function to obtain the apparent rate of prolyl isomerization, which then was plotted as a function of the enzyme concentration and fitted according to Eq. (6), where the slope is equal to the catalytic efficiency[36].

$$k_{app} = k_0 + [E] \cdot \frac{k_{cat}}{K_M},$$  (6)

where $k_{app}$ is the measured rate of isomerization in $s^{-1}$, $k_0$ the uncatalyzed rate in $s^{-1}$, [E] the enzyme concentration in nM and $k_{cat}/K_M$ the catalytic efficiency ($nM^{-1} s^{-1}$).

For enzyme inhibition experiments 12 nM CypD and increasing concentrations of CsA (100 μM–1 mM stock in DMSO, Sigma) were added to the cuvette and incubated for 2 min at room temperature in 10 mM Hepes, 140 mM KNO₃, pH 7.4. Liposomes, prepared as for the stopped-flow assay but without ANTS, were added for the control experiment in Supplementary Fig. 3c. The reaction was started by adding Abz-Ala-Leu-Pro-Phe-pNA to a final concentration of 4 μM and monitoring the fluorescence increase. Traces were fitted to a mono-exponential function to obtain the apparent rate of prolyl isomerization. This rate was plotted as function of the CsA concentration and analyzed according to Eq. (5), The highest concentration of CsA used in these experiments was 10 μM due to the low solubility of CsA in aqueous buffer.

**Sample preparation for cryoEM studies.** In order to prepare samples for structural studies using cryoEM[13], the final gel filtration during protein purification was performed in 20 mM Hepes, 100 mM KCl, 0.5 mM DDM, 200 μM cAMP, pH 7.4. SthK P300A was then reconstituted into lipid nanodiscs by mixing SthK P300A with MSP1E3 (Addgene #20064)[64] and POPG in a molar ratio of 1:1:70 and incubated for 1 h. Detergent removal was initiated by adding 20 μg SM-2 BioBeads (BioRad) per 100 μl of sample. The sample was then incubated for 2 h at 4 °C under constant agitation, BioBeads were changed and the sample was incubated overnight at 4 °C. The nanodisc-containing supernatant was cleared by centrifugation and applied to a Superose 6 10/300 (GE Lifesciences) gel filtration column equilibrated in 10 mM Hepes, 100 mM KCl, 200 μM cAMP, pH 7.4. The peak corresponding to SthK P300A incorporated in nanodiscs was collected and concentrated to 7.5 mg/ml (assuming that 1 A₂₈₀ = 1 mg protein, Supplementary Fig. 5a). cAMP was added to the final sample to reach a concentration of 3 mM.

Grids for cryoEM were prepared and frozen as described[13]. Right before freezing, the sample was supplemented with 3 mM fluorinated Fos-choline 8 (Anatrace). 3.5 μl of the sample were applied to a glow-discharged gold grid (UltrAu-Foil R1.2/1.3 300-mesh, Quantifoil) and incubated for 10 s. A Vitrobot Mark IV (FEI) was used to blot the excess sample with a blot force of 0 and 2 s blot time at 22 °C and 100% humidity before plunge-freezing in liquid ethane.

**CryoEM data collection.** Automated data collection using Leginon[65] was carried out at a Titan Krios (FEI) operated at 300 kV equipped with a Quantum GIF and a K2 direct electron detector (Gatan, slit width 20 eV) and a Cs corrector. The calibrated pixel size was 1.0961 Å/pixel with a nominal defocus of −1 to −2.2 μm. 2494 movies (50 frames each) were acquired over an exposure time of 10 s at a total dose of 71 $e^-/Å^2$ (1.4 $e^-/Å^2$/frame).

**Processing of cryoEM data.** The processing of cryoEM data was performed in Relion3.0 beta[66]. Movie stacks were imported into Relion, motion corrected using Relion's implementation of MotionCor2, followed by CTF-estimation on the dose-weighted micrographs (using CTFFIND4[67]). Template-based particle picking was performed using the published density for SthK (EMDB: 7484[13]) with a 25 Å low-pass filter (Supplementary Fig. 5b) and the resulting 657943 particles were binned two-times. After two rounds of 2D classification (Supplementary Fig. 5c), the selected 418649 particles were re-extracted without binning (box size 256 px) and initial processing of the data in 3D was performed without symmetry. The obtained density was four-fold symmetric similar to our previous structure of WT SthK[13]. Subsequently, particles were 3D classified into 10 classes with C4 symmetry, to achieve the highest possible homogeneity in the sample. A soft mask was applied during 3D classification that included portions of the nanodisc to ensure correct particle orientation (Supplementary Fig. 5d, e). All 10 classes were refined separately with C4 symmetry. Two different states were identified after 3D refinement with global angular sampling of 7.5 degrees (210683 particles and 19918 per class). The densities of both states could be used to generate soft masks and the refinements were continued using solvent flattened FSCs. The same masks were used to perform 3D classification without alignment to sort particles according to their resolution (Supplementary Fig. 5 f,g). The best class for each state was re-refined (7.5 degree global angular sampling) and refinement was continued using a soft mask. Multiple rounds of CTF-refinement, Bayesian polishing, 3D auto refinement and postprocessing were performed until the final resolution converged (3.4 Å resolution for the higher-populated state). The local resolution was calculated in Relion3 beta (Supplementary Figs. 5, 6).

For the putatively open state, the initial map was first used to generate a poly-Ala model using Phenix[68]. Real-space refinement[69] was performed using the cAMP-bound structure of SthK (PDB: 6CJU[13]) as starting model. All side chains were reduced to Alanine before the refinement and strong main chain and secondary structure restraints were applied. Initially, 20 iterations were performed

with morphing and simulated annealing during each cycle. The initial molecular model was then used to create a mask for signal subtraction in Relion3. Particles were refined and the density was improved by applying a soft mask (6.7 Å final resolution, Supplementary Fig. 6b).

**Model building.** The model for cAMP-SthK (PDB: 6CJU[13]) was used as starting model to build the structure for SthK P300A in the closed state. Refinement was performed using Phenix and the structure was subjected to multiple rounds of refinement including morphing and simulated annealing[68,69]. The resulting model was manually optimized in Coot[70], cAMP and POPG lipid molecules were placed in the corresponding densities before final model optimization in Phenix.

The putatively active state, with low-resolution map, was modeled using the closed-state structure of SthK P300A as starting model. Multiple rounds of refinement were performed with morphing and simulated annealing to fit the model into the density while applying strong secondary structure restraints. The density for the first transmembrane helices S1–S4 was not completely resolved and thus strong main chain and secondary structure restraints were necessary. Placement of the model inside of the density was confirmed in Coot. No side chains could be assigned, and all side chains were reduced to Ala. The protein register was maintained from the closed-state reference structure.

Both models were validated using Phenix[71,72]. First, the FSC between the final refined model and the final map was calculated (FSC_sum). Random shifts of 0.3 Å were introduced in the final model. This modified model was then refined against one of the unfiltered half maps, and the FSC was calculated (FSC_work). The newly refined model was then used to calculate the FSC between this model and the second half map (FSC_free), which had not been used for refinement. The final reports are presented in Supplementary Table 2. The similarity between these FSC values indicates that there was no overfitting (Supplementary Fig. 6c–f).

**Kinetic simulations of multi-state mechanisms.** We performed kinetic simulations, using QUB 2.0.0.34[73], to calculate the response of an ensemble of 10000 channels to cAMP application (our stopped-flow experiments are equivalent to a macroscopic current in response to fast ligand application). Transitions between states are described by microscopic rate constants. Transitions associated with ligand binding are dependent on the cAMP concentration ($k_{on} \cdot$ [cAMP]). Since our experimental results do not provide information about the microscopic rate constants, we did not attempt to reproduce the exact results. We rather adjusted the values of the rates (see Supplementary Table 3) to qualitatively fulfill the following requirements: (1) the activation current is bi-phasic and the two time constants are roughly in the millisecond and second time range, respectively, (2) single-channel gating occurs on the millisecond time scale with low maximal open probability to match the experimental data[11], (3) the EC₅₀ of the cAMP dose responses qualitatively match the values and the shifts observed in the experiments. For the first requirement, we simulate the current response after a concentration jump, using the scalar expression of the current as a weighted sum of k-1 exponential components[74,75] (Eq. (7)).

$$I(t) = I(eq) + \sum_{i=2}^{i=k} b_i \cdot e^{-\frac{t}{\tau_i}}$$  (7)

with $I(t)$ being the macroscopic current at time $t$, $I(eq)$ the macroscopic current at equilibrium, $b_i$ the coefficients that define the amplitude for each component $i$, and $\tau_i$ the time constants of the exponential components. Parameters $b_i$ and $\tau_i$ are complex functions of all rate constants in the model as well as, for $b_i$, of the initial conditions[74,75].

For the second requirement, the rates for the open-closed equilibrium were fixed to yield ms openings in simulated single-channel recordings with a Po of ~0.1[11]. For the third requirement, current amplitudes are calculated from simulations at different cAMP concentrations and plotted to determine the EC₅₀ values according to Eq. (4). The *cis/trans* equilibrium for apo SthK was determined by setting the ligand concentration to 0 μM and used as the starting point for all simulations. The mechanism presented in Fig. 3c and Fig. 5a enables *cis/trans* prolyl isomerization in all three states (apo, cAMP-bound, open). Efficient catalysis by PPIases suggests, that Pro300 in SthK is fully accessible to these enzymes and isomerization likely can happen in all three states (Fig. 3c). However, in order to reduce the number of parameters, we omitted isomerization in the cAMP-bound state for our simulations (mechanism presented in Fig. 5a). The inclusion of transitions between states 3 and 4 employing the same rates as for the 5 ↔ 6 transitions, did not change the outcome of the simulations. Since states 3 to 6 are all cAMP-bound, we think it is reasonable to assume that isomerization rates between 3 and 4 and 5 and 6 would be similar.

**Reporting summary.** Further information on research design is available in the Nature Research Reporting Summary linked to this article.

## Data availability

Data supporting the findings of this manuscript are available from the corresponding author upon reasonable request. A reporting summary for this article is available as a Supplementary Information file. The maps for SthK P300A in the closed-state and the

putatively open state have been deposited in the Electron Microscopy Data Bank (EMDB) under accession codes 21453 and 21454, respectively. Atomic coordinates for closed-state SthK P300A and putatively open SthK P300A have been deposited in the Protein Data Bank (PDB) with accession codes 6VXZ and 6VY0, respectively. Structures produced for this study were compared to previously published structures of apo WT SthK (PDB: 6CJQ, cAMP-bound WT SthK (PDB: 6CJU) as well as the cryoEM density for cAMP-bound WT SthK EMD-7484. Data were also compared to the isolated C-linker/CNBD of SthK bound to cAMP (PDB: 4D7T). Source data are provided with this paper.

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

## Acknowledgements

The authors thank M. Seeliger and G. Zoldak for providing us with the genes for CypD and SlyD, respectively. Tetra-peptides to test PPIase activity were a gift from F. Schmid. We thank A. Accardi and F. Schmid for critically reading the manuscript. CryoEM data collection was performed at the Simons Electron Microscopy Center and National Resource for Automated Molecular Microscopy located at the New York Structural Biology Center, supported by grants from the Simons Foundation (SF349247), NYSTAR, and the NIH National Institute of General Medical Sciences (GM103310). The work presented here was sponsored by the NIH (GM124451 to CN) and the American Heart Association (18POST33960309 to PS).

## Author contributions

P.S. and C.N. designed research. P.S. performed all experiments and analyzed the data. J. R. froze the grid for cryoEM, collected the cryoEM data and provided initial guidance for cryoEM data analysis. P.S. performed all cryoEM data processing and model building. P.S. and C.N. interpreted the results, wrote and revised the manuscript.

## Competing interests

The authors declare no competing interests.
