## [Peer Review File · Nature Communications]

REVIEWER COMMENTS

Reviewer #1 (Remarks to the Author):

This is an interesting study that investigates the functional role of prolyl cis-trans isomerization in the activation kinetics of a cAMP-gated ion channel. The study focuses on SthK, a bacterial homolog of human hyperpolarization-activated channels (HCN, or pacemaker channels) that play a critical role in regulating heart rate by resetting the action potential. SthK previously was shown to display slow biphasic activation kinetics with time constants that differ by two orders of magnitude (20 ms and 2 s). In the current manuscript, a conserved proline residue (Pro300 in SthK) in a loop within the cytosolic cyclic nucleotide-binding domain (CNBD) is shown by mutation and by isomerase-catalyzed enzymatic assays to control the activation kinetics of SthK. The authors conclude that the trans and cis isomers of the Val299-Pro300 peptide bond give rise to the 20 ms and 2s activation time constants, respectively, and that this isomerization serves as a regulatory mechanism for SthK and by analogy, possibly for human HCN channels. This study, which reveals a novel role of prolyl cis-trans isomerization in the activation of this bacterial channel, should have significant impact and broad interest in the scientific community since it has strong implications for how this molecular switch is potentially involved in setting the pace of the human heart beat.

The approach is thorough with multiple angles of characterization, and the conclusions are supported by the data. Briefly, the authors showed that mutation of Pro300 to Ala (P300A) or Val (P300V), either of which abolishes the cis isoform, resulted in channel activation within the instrument dead time (12 ms) but with similar single-channel ion conductance amplitudes and open probabilities compared to WT. The addition of a prolyl isomerase enzyme (either a FKBP or a cyclophilin) abolished the slow (2s) activation, while further addition of isomerase inhibitor cyclosporine A reversed this effect, supporting the role of cis-trans isomerization in channel activation. In agreement with classical enzyme behavior, the addition of isomerase was shown to not alter the equilibrium populations of open and closed channel states. It was found that the affinity for cAMP changes over time, consistent with assignment of the faster and slower activation states to Pro300 trans and cis states, respectively. Finally, cryo-EM structural models of P300A SthK (major and minor populations) were constructed and compared to the previously determined WT structure. This is an important set of experiments that, when taken together, support the conclusions.

All previously conveyed questions and concerns from this reviewer have been thoroughly addressed by the authors in their revised manuscript. This manuscript is an important addition to the set of examples in which prolyl cis/trans isomerization acts as a timing device in complex biological processes.

Reviewer #2 (Remarks to the Author):

We appreciate the authors' efforts to address our major points through further explanations of their original data and literature. We think their answers to point 1 and 2 about the significance of their findings are satisfactory and seem to address most of our concerns. Point 4 about the overly speculative discussion is also well addressed. However, for the remaining points (3, 5-8), the authors comment that they have no means to answer the questions we asked or provide primarily speculative answers without any additional data. For example, the authors respond to point 3, but a structure of the apo WT SthK channel showing the cis and trans isomers is still lacking. Another example is point 5. The author's answer to this point assumes cAMP cannot penetrate within the liposomes. There are several cell permeable cAMP analogs that could be used as controls to check if there is a difference compared to cAMP. The authors also rule out the use of whole cell patch-clamp because of the presence of cellular PPIases. This is a good point, but such PPIase could be inhibited or outcompeted by using Pro containing peptides or analogs. No attempt was made to actually add some of these

controls. In conclusion, while we think this new version of the manuscript is significantly improved compared to the original manuscript, especially with respect to the significance, room is still left to further strengthen this interesting story through further control experiments.

Reviewer #3 (Remarks to the Author):

In this work the authors characterize the slow cAMP-induced activation of SthK channels and the mechanism underlying the channels bi-phasic activation characteristic. They showed that a cis/trans isomerization of proline 300 modifies SthK cAMP affinity, dictating the faster/slower activation, which can be modified by Prolyl isomerases. This is a very interesting and underexplored mechanism for ion channel modulations and the potential generalization to HCN channels make this manuscript of significant interest.

I think the manuscript reads well and that the data supports the conclusions made. I did focus my review on the stopped-flow experiments and overall found them to be well done and presented. I did find a few points that can use calcification and also made a few suggestions that might improve the manuscript.

Page 5, "within the dead time of the instrument (12 ms, Figure 1E, F)". I was very confused when I read this as I remember the dead time of the SX.20 stopped-flow to be around one ms, until I realized you are referring to the minimum pre-mix / aging time, please make the wording unambiguous.

Page 7, "Application of either SlyD or CypD to the SthK-containing liposomes in the stopped-flow assay abolished the slow activation of SthK, with the channels displaying maximal activity after the shortest mixing time, ...". When I first read this one, I mistakenly took it that SlyD or CypD were added in the stopped-flow (in the pre missing step) and not 10 min before as mentioned in the methods, which makes much more sense. Maybe a tiny rewording can avoid that confusion for others.

Page 17, "The entire reconstitution was performed in the absence of cAMP to silence channels that incorporate with their CNBDs in the lumen of the vesicles. Accordingly, only channels that face the outside of the vesicles with their cytosolic domains can be activated by externally applied cAMP and are substrates for PPIases." I am assuming cAMP crosses the bilayer extremely slowly compared to the mixing times used here but that is not mentioned.

Page 18, "During one experiment, at least seven repeats were performed for each data point, every repeat was visually inspected for its quality, mixing artifacts were sorted out and the remaining repeats were analyzed". Of the seven technical repeats approximately how many are "normally" removed/remain?

Both stopped-flow and stopped flow used a few times.

In the stopped-flow methods: how long of a stretch was the stretch exponential fitted to? Full 3 s or only initial as you are capturing the initial rate (at 2 ms), and what sampling rate did you use for the stopped-flow data?

Fig. 1, 2, S2, S4. Are the stopped-flow fluorescence curves, representative samples and averages of those technical repeats or representative technical trace from a representative sample?

Fig. 2E, mixing time is not mentioned in caption, I found it in the methods 12 ms, but as it's mentioned for all other panels in the caption might be good to also add for this one.

Fig. S1 TI+ "it cannot enter through closed channels and the leak across the membrane is orders of magnitude slower than through activated channels." I see the no cAMP close channel controls showing

quite slow but still some reduction in fluorescence, through the membrane only would always be slower than that it would still be interesting to see a lipid vesicle only TI+ leakage control, or reference to expected leakage / fluorescence drop value.

TI+ leakage controls for the different added substrates. I was happy to see that the authors included a number of those (0 cAMP in Fig. 1, S2 and S4) for most substrates but they are never called out as such. Minor point as they all appear very flat but would be good to mention them explicitly and their affect (lack of) on the rate (maybe in SI).

For the stopped-flow normalized rate numbers per sample, provided in the accompanying excel file. I was surprised so see a number of values as exactly 1 (and quite some even 1.0000). What is the reason for that?

Reviewer #1 (Remarks to the Author):

This is an interesting study that investigates the functional role of prolyl cis-trans isomerization in the activation kinetics of a cAMP-gated ion channel. The study focuses on SthK, a bacterial homolog of human hyperpolarization-activated channels (HCN, or pacemaker channels) that play a critical role in regulating heart rate by resetting the action potential. SthK previously was shown to display slow biphasic activation kinetics with time constants that differ by two orders of magnitude (20 ms and 2 s). In the current manuscript, a conserved proline residue (Pro300 in SthK) in a loop within the cytosolic cyclic nucleotide-binding domain (CNBD) is shown by mutation and by isomerase-catalyzed enzymatic assays to control the activation kinetics of SthK. The authors conclude that the trans and cis isomers of the Val299-Pro300 peptide bond give rise to the 20 ms and 2s activation time constants, respectively, and that this isomerization serves as a regulatory mechanism for SthK and by analogy, possibly for human HCN channels. This study, which reveals a novel role of prolyl cis-trans isomerization in the activation of this bacterial channel, should have significant impact and broad interest in the scientific community since it has strong implications for how this molecular switch is potentially involved in setting the pace of the human heart beat.

The approach is thorough with multiple angles of characterization, and the conclusions are supported by the data. Briefly, the authors showed that mutation of Pro300 to Ala (P300A) or Val (P300V), either of which abolishes the cis isoform, resulted in channel activation within the instrument dead time (12 ms) but with similar single-channel ion conductance amplitudes and open probabilities compared to WT. The addition of a prolyl isomerase enzyme (either a FKBP or a cyclophilin) abolished the slow (2s) activation, while further addition of isomerase inhibitor cyclosporine A reversed this effect, supporting the role of cis-trans isomerization in channel activation. In agreement with classical enzyme behavior, the addition of isomerase was shown to not alter the equilibrium populations of open and closed channel states. It was found that the affinity for cAMP changes over time, consistent with assignment of the faster and slower activation states to Pro300 trans and cis states, respectively. Finally, cryo-EM structural models of P300A SthK (major and minor populations) were constructed and compared to the previously determined WT structure. This is an important set of experiments that, when taken together, support the conclusions.

All previously conveyed questions and concerns from this reviewer have been thoroughly addressed by the authors in their revised manuscript. This manuscript is an important addition to the set of examples in which prolyl cis/trans isomerization acts as a timing device in complex biological processes.

We would like to thank the reviewer again for their insightful comments that helped to improve our manuscript.

Reviewer #2 (Remarks to the Author):

We appreciate the authors' efforts to address our major points through further explanations of their original data and literature. We think their answers to point 1 and 2 about the significance of their findings are satisfactory and seem to address most of our concerns. Point 4 about the overly speculative discussion is also well addressed. However, for the remaining points (3, 5-8), the authors comment that they have no means to answer the questions we asked or provide primarily speculative answers without any additional data. For example, the authors respond to point 3, but a structure of the apo WT SthK channel showing the cis and trans isomers is still lacking. Another

example is point 5. The author's answer to this point assumes cAMP cannot penetrate within the liposomes. These are several cell permeable cAMP analogs that could be used as controls to check if there is a difference compared to cAMP. The authors also rule out the use of whole cell patch-clamp because of the presence of cellular PPlases. This is a good point, but such PPlase could be inhibited or outcompeted by using Pro containing peptides or analogs. No attempt was made to actually add some of these controls. In conclusion, while we think this new version of the manuscript is significantly improved compared to the original manuscript, especially with respect to the significance, room is still left to further strengthen this interesting story through further control experiments.

We would like to thank the reviewer for the assessment that the question about significance of our work was addressed appropriately. As for the remaining points, we seem to have misunderstood some of them as the new comments from the reviewer suggest, and we hope we addressed them better now, see below.

Original Point 3: "A structure of the apo WT SthK channel showing the *cis* and *trans* isomers would be more helpful than the structure of cAMP-bound P300A SthK."

Current Point 3: "For example, the authors respond to point 3, but a structure of the apo WT SthK channel showing the *cis* and *trans* isomers is still lacking."

Yes, unfortunately the structure of apo WT SthK channel showing the *cis* and *trans* isomers at Pro300 is not possible, as much as we would want it. In brief, a resolution of ~ 2.5 Å or better would be necessary in the siphon region to unambiguously assign the backbone angles of the protein chain, and this is not the case here. Our best WT SthK structure was solved to an overall resolution of 3.3 Å¹ (WT SthK bound to cAMP), and the siphon region, where Pro300 is located, is resolved with a resolution of ~ 4 Å (Revision Fig 1, middle). However, we did go back to our WT SthK structures and carefully checked the local resolution. As shown in Revision Fig 1 below, where we plotted the structures of apo WT SthK, cAMP WT SthK, and SthK P300A zoomed into the CNBDs colored based on local resolution, the resolution of the siphon loop (black square) is markedly lower than that in the rest of the CNBD, similar to that of the C-helix (labeled as α C in the figure), which is the other domain known to move during gating. These regions are thus some of the most flexible in the channel. For the siphon, we propose that this is mainly due to the existence of 2 different configurations of the Pro300 (*cis* and *trans*). In a case where two structures in the same sample are only different at a single proline (*cis* or *trans*), classification algorithms used in cryoEM analysis are currently not able to separate the *cis* and *trans* species given the small, structural difference, and thus, only one structure, averaged over the two Pro configurations, and as such, with lower local resolution, will be obtained (see new text on page 10). We extended the panel in Fig S8 (S8E), that was added for the previous round of reviews, highlighting this. The panel now shows the local resolution of the entire CNBDs as well as a zoom into only the siphon region of apo WT, cAMP-bound WT, and SthK P300A.

Revision Fig 1: Local resolution of the C-linker/CNBD as calculated using Relion 3 for apo WT SthK (EMD-7482¹), cAMP-bound WT SthK (EMD-7484¹), and cAMP-bound SthK P300A. All three densities are colored according to the same coloring scheme and the siphon is highlighted by a black square.

Original comment 5a: *“Is there any chance that SthK enters the liposome inside out with CNBD within the liposomes? If so, how are SlyD or CypD injected inside the liposomes? What is the effective concentration of SlyD or CypD inside the liposomes?”*

Our answer to this question was that SthK will likely insert either way into liposomes but that this does not matter in our setup because the channels inserted with the CNBDs inside are inactive since we have 0 cAMP inside the liposomes. Only the channels oriented with the CNBDs outside of liposomes are exposed to fast cAMP application and are substrates for SlyD and CypD. Our experiments only assay the SthK channels oriented with CNBD outside, while the channels oriented with CNBDs inside are silent because they are exposed to 0 cAMP. These experiments were specifically designed to account for unknown orientation of the channel.

Current comment 5a: *“Another example is point 5. The author’s answer to this point assumes cAMP cannot penetrate within the liposomes. These are several cell permeable cAMP analogs that could be used as controls to check if there is a difference compared to cAMP.”*

Only based on this new comment, it became clear that the reviewer expected more from us. We originally believed that the reviewer requested clarification about our liposome reconstitution and experimental setup rather than asking for new experiments. In the current comment, the reviewer suggests that we should compare our results with new experiments where instead of regular cAMP, we should use cell-permeable cAMP analogs. The reviewer calls the experiment a control but did not explain what it would control for. We believe, and please correct us if we are wrong, that the reviewer suspects that the slower (2 s) activation component we observe could be an artifact due to leakage of cAMP through the bilayer which could lead to slowly activating SthK channels with the CNBDs facing the inside of the liposome. This is not the case for a number of reasons. First, cAMP is well-known not to be membrane permeable, with a $\log P = -3.4$ as calculated with ChemAxon². In addition, even the permeation of “membrane-permeant cAMP analogs” across membranes is much slower than the slow component we observe here³⁻⁵. Experiments using such analogs are performed by incubating cells for at least 30 min with these analogs before measurements. Second, and most important, we already have built-in controls for this potential issue: if the slow phase was simply due to cAMP leaking and activating the intravesicular-facing CNBD channels, then the mutants P300A and P300V SthK should also display such a slow component, since the experiments are performed in exactly the same way. However, as observed in Fig 1E,F and S2D,E these channels activate very fast, within the mixing time of the setup, with no slow component. These controls strongly argue against any cAMP leakage across the bilayer (we added a statement to that effect in the legend of Supplementary Fig S2).

Original comment 5b: *“Would the effect of CypD or SthK still be observed using whole-cell patch-clamp and cells overexpressing the ion channel?”*

We argued that whole cell patch-clamp will not be able to resolve the kinetics, because of the presence of cellular PPlases, undefined cellular cAMP levels, and, most importantly, in whole-cell patch-clamp experiments the inside is not accessible for rapid perfusion with ligand. To reliably analyze this sophisticated regulation, it is also important to start from clean, homogeneous, apo protein.

Current comment 5b: *“The authors also rule out the use of whole cell patch-clamp because of the presence of cellular PPlases. This is a good point, but such PPlase could be inhibited or outcompeted by using Pro containing peptides or analogs. No attempt was made to actually add some of these controls”*

Indeed, cellular PPlases can be blocked or inhibited, but this also would inhibit the acceleration of the slow activation step by intentionally applied isomerases, a key element of our study.

Importantly, whole-cell patch-clamp still will not allow rapid perfusion with ligand, and still have undefined cellular cAMP levels, as well as difficulty in controlling the channel's liganded state. In summary, all these limitations make whole cell patch-clamp not suitable to study prolyl isomerization.

It is important to keep in mind that this is really a pioneering study. It presents rigorous work performed to address that this enzymatic regulation really occurs in this particular ion channel, SthK. One of the key advantages of using prokaryotic channels, which people sometimes overlook, is that they can be used as models to properly dissect a mechanism. This is because one can express and purify large quantities of stable, functional protein and reconstitute them in minimalistic systems, thus allowing to address specific questions without potential interference. In our experiments we know exactly where the observed channel activity comes from. We know exactly which PPIase is present and at what concentration because we added the purified enzyme. Only such a controlled and defined environment allowed us to detect and characterize the effects of regulatory prolyl isomerization. In contrast, the function of eukaryotic proteins often depends on lipid molecules in trace abundance and other cellular proteins that are difficult to control. Thus, a prokaryotic system can be superior towards establishing a mechanism compared to the eukaryotic counterpart. We would like to stress again that prolyl isomerization intrinsically is a kinetic process and careful, tightly controlled kinetic experiments, such as those presented in this manuscript, are crucial.

Original comment 6: "*How do the P300A, SlyD or CypD affect the kinetics of cAMP binding (on/off rates)?*"

Our answer was that this is a very interesting question that we are also interested in, but that we currently have no assay to analyze the binding kinetics of cAMP to SthK.

Current comment 6: "*However, for the remaining points (3, 5-8), the authors comment that they have no means to answer the questions we asked or provide primarily speculative answers without any additional data.*"

The reviewer wants us to perform kinetic binding experiments to determine the on and off binding-rates of cAMP to the WT channel with and without isomerases, and the P300A mutant. In our model shown in Figure 3C and 5A (and Revision Fig 2), we indeed only model these rate constants for the *cis* and *trans* Pro species of the channel. We agree that it would be nice to be able to constrain these rate constants for the binding steps but, as the reviewer is probably aware, such experiments have been shown to not only be extremely challenging but also strongly dependent on the model of ligand binding and gating used for analysis^{6,7}. For example, any measurement of a binding rate for a ligand that when it binds also activates the channel, would not give the clean intrinsic binding rate, but rather a value that reflects a mix of the binding rate to the open channel, the binding rate to the closed channel, weighted by the fractions of channels existing in the different states, hence its dependence on a model for channel gating. In our case, the mentioned open/closed equilibrium additionally is coupled to the *cis/trans* equilibrium at Pro300 adding even more complexity (Revision Fig 2). Thus, a kinetic measurement, that would reveal a 2-(or more) component ligand binding step, would almost certainly reflect multiple layers of complexity rather than simply the two binding components for a *cis* and *trans* Pro-containing channel. For these reasons, we consider such experiments to be beyond the scope of this current manuscript.

Revision Fig 2: panel A of Figure 5 from the presented manuscript is shown. Model of the modulatory prolyl isomerization in SthK used for simulations. Transitions in grey were omitted for the simulations since their inclusion does not change the outcome and to reduce the number of parameters.

Original comment 7: “*Moroni et al. have previously shown that cAMP affects the oligomerization states of the intracellular region of HCN2 and 4. Will the oligomerization states affect the Cis Trans equilibrium? Or is the Cis Trans equilibrium affected by the oligomeric states?*”

We understood this question as the reviewer inviting us to speculate on how would different oligomerization states observed in a soluble protein composed of just the intracellular regions of HCN1, 2, 4 channels may translate into a phenotype in the full-length SthK channels and how would this relate to our described channel regulation modality. In our reading, there was no experiment suggested or requested. We responded by providing a speculation in the discussion section.

Current comment 7: “*However, for the remaining points (3, 5-8), the authors comment that they have no means to answer the questions we asked or provide primarily speculative answers without any additional data.*”

In light of the reviewer’s concern, we cloned, expressed and purified the C-linker/CNBD domain of SthK in the absence of cAMP and subjected this protein to analytical gel filtration in the absence and presence of added cAMP, in order to test whether oligomerization changes upon liganding, similar to the experiment by Lolicato *et al*⁸ mentioned by the reviewer. In Lolicato *et al*⁸, the intracellular region of HCN channels forms a tetramer in the presence of cAMP and collapses into monomers (to varying extents) in the absence of cAMP. However, in our experiments, the equivalent region in SthK appears predominantly tetrameric in both the presence and the absence of ligand (major peak at 1.8 ml in Revision Fig 3), suggesting that the tetramer is more robust in SthK than in some HCN isoforms. Since the oligomerization state does not change significantly in this case, the reviewer’s question does not appear applicable to SthK.

Revision Fig 3: Gel filtration profile of the purified C-linker/CNBD of SthK as monitored by protein fluorescence (excitation 280 nm, emission 330 nm) on a Superdex 200 5/150 column (GE Lifesciences) at a flow rate of 0.3 ml/min. 10 μl of a 10 μM protein solution were injected. Samples were prepared in 20 mM Hepes, 100 mM KCl, pH 7.4, ± 200 μM cAMP and gel filtration was performed in the same buffers. Elution profiles for the C-linker/CNBD of SthK in the absence (blue) and presence (black) of 200 μM cAMP are shown.

Original comment 8: “8. *If the channel forms tetramers, are all four apo CNBDs with a Pro300 Cis conformation? Or could there be a mixture of Cis and Trans? Can we explain this case with Eq. 7?*”

We took the original question as a request to expand and speculate on this topic. Our response was that our stopped-flow assay is a macroscopic recording that cannot provide the resolution to distinguish between single CNBDs within tetrameric channels and we could not speculate about the number of subunits with *cis* Pro300 needed to establish slow activation or if the concerted activation mechanism also applies to the conformational state at Pro300 (which would mean all-*trans* or all-*cis*). Without this piece of information, Eq 7 is not able to differentiate between the two cases, as it only refers to the percentage of *cis* and *trans* Pro300 in the entire assay volume. We commented on this limitation in the Results section where we present the time-dependent shift of the EC₅₀ value in WT SthK.

Current comment 8: “*However, for the remaining points (3, 5-8), the authors comment that they have no means to answer the questions we asked or provide primarily speculative answers without any additional data.*”

We cannot think of a realistic experiment that would lead to the determination of the conformational state (*cis* or *trans*) at Pro300 at each individual subunit within the tetramer. Theoretically, a cryo-EM experiment could be proposed for something like this, and if the resolution (without imposing symmetry) and quality is high in the regions in question, such a determination could be made. However, as we pointed out in our response to point 3 and in the new panel added to Supplementary Fig S8, the resolution in the siphon containing Pro300 is low, likely due to flexibility in this region. We believe that this indeed indicates heterogeneity in this region pointing to the co-existence of *cis* and *trans* Pro300. Unfortunately, we cannot separate classes that would correspond to these two configurations.

“No attempt was made to actually add some of these controls. In conclusion, while we think this new version of the manuscript is significantly improved compared to the original manuscript, especially with respect to the significance, room is still left to further strengthen this interesting story through further control experiments.”

We appreciate that the reviewer considers our story interesting and we hope that the changes we made in this newly revised version of the manuscript, as well as the experiments we performed in order to address some of the reviewer’s remaining questions are satisfying. As we hope we explained in detail above, we did not interpret the reviewer’s original comments and questions (5-8) as suggestions for additional experiments or controls.

Reviewer #3 (Remarks to the Author):

In this work the authors characterize the slow cAMP-induced activation of SthK channels and the mechanism underlying the channels bi-phasic activation characteristic. They showed that a cis/trans isomerization of proline 300 modifies SthK cAMP affinity, dictating the faster/slower activation, which can be modified by Prolyl isomerases. This is a very interesting and underexplored mechanism for ion channel modulations and the potential generalization to HCN channels make this manuscript of significant interest.

I think the manuscript reads well and that the data supports the conclusions made. I did focus my review on the stopped-flow experiments and overall found them to be well done and presented. I did find a few points that can use calcification and also made a few suggestions that might improve the manuscript.

We would like to thank the reviewer for carefully reading our manuscript and the insightful comments that will help to make our work clearer to the reader.

Page 5, “within the dead time of the instrument (12 ms, Figure 1E, F)”. I was very confused when I read this as I remember the dead time of the SX.20 stopped-flow to be around one ms, until I realized you are referring to the minimum pre-mix / aging time, please make the wording unambiguous.

The reviewer is correct and points out an ambiguity, which we now clarified. We updated the text on page 4: ... activation of SthK P300A is fast, and complete after the shortest mixing time (12 ms, Figure 1E, F).

Page 7, “Application of either SlyD or CypD to the SthK-containing liposomes in the stopped-flow assay abolished the slow activation of SthK, with the channels displaying maximal activity after the shortest mixing time, ...”. When I first read this one, I mistakenly took it that SlyD or CypD were added in the stopped-flow (in the pre mixing step) and not 10 min before as mentioned in the methods, which makes much more sense. Maybe a tiny rewording can avoid that confusion for others.

We changed the sentence on page 7 to: *In the presence of either SlyD or CypD, the slow activation phase of SthK was abolished, and the channels displayed maximal activity after the shortest mixing time.*

Page 17, “The entire reconstitution was performed in the absence of cAMP to silence channels that incorporate with their CNBDs in the lumen of the vesicles. Accordingly, only channels that face the outside of the vesicles with their cytosolic domains can be activated by externally applied cAMP and are substrates for PPLases.” I am assuming cAMP crosses the bilayer extremely slowly compared to the mixing times used here but that is not mentioned.

The calculated logP value for cAMP is -3.4 (using ChemAxon ²), meaning that cAMP prefers aqueous environment over hydrophobic environments by a factor of ~2500, indicating that cAMP is not crossing the membrane at appreciable rates during the experimental time. Consequently, cell-based studies employ membrane-permeant cAMP analogs ³⁻⁵. We updated the methods on page 17: *Accordingly, only channels that face the outside of the vesicles with their cytosolic domains can be activated by externally applied cAMP (cAMP is not membrane permeable, logP = -3.4 as calculated according to ²) and are substrates for PPLases.*

Page 18, “During one experiment, at least seven repeats were performed for each data point, every repeat was visually inspected for its quality, mixing artifacts were sorted out and the remaining repeats were analyzed”. Of the seven technical repeats approximately how many are “normally” removed/remain?

At least seven repeats were performed after the flow circuit was equilibrated with the respective solution. Only in rare cases (<5 %) did we remove two repeats. In these cases, the minimum number of technical repeats for a data point within one experimental repeat is 5. “Normally” 6-7 repeats were used for the analysis. We updated the methods on page 19: ... *and the remaining repeats (typically 6-7 technical repeats) were analyzed separately...*

Both stopped-flow and stopped flow used a few times.

Thank you. We updated the manuscript and now consistently use stopped-flow.

In the stopped-flow methods: how long of a stretch was the stretch exponential fitted to? Full 3 s or only initial as you are capturing the initial rate (at 2 ms), and what sampling rate did you use for the stopped-flow data?

Only the first 100 ms were used to fit in order to obtain the initial rate of Tl^+ influx (at 2 ms) as the reviewer states correctly. 5000 data points were recorded over the time of 1 s. We added both details in the methods section page 18: *5000 data points were recorded over 1 s, however, to obtain the initial rate of Tl^+ influx only the first 100 ms were analyzed.*

Fig. 1, 2, S2, S4. Are the stopped-flow fluorescence curves, representative samples and averages of those technical repeats or representative technical trace from a representative sample?

All traces presented in this manuscript are averaged over at least five technical repeats from one sample. In response to the reviewer, we added a note in the methods section page 18: *All presented quenching kinetics are averaged over at least five technical repeats of a representative experiment.*

Fig. 2E, mixing time is not mentioned in caption, I found it in the methods 12 ms, but as it's mentioned for all other panels in the caption might be good to also add for this one.

We thank the reviewer for pointing out our omission. We added the mixing time in the legend of Figure 2 since it is an important experimental information.

Fig. S1 Tl^+ "it cannot enter through closed channels and the leak across the membrane is orders of magnitude slower than through activated channels." I see the no cAMP close channel controls showing quite slow but still some reduction in fluorescence, through the membrane only would always be slower than that it would still be interesting to see a lipid vesicle only Tl^+ leakage control, or reference to expected leakage / fluorescence drop value.

Thank you for this suggestion. We now include a trace of protein free liposomes in Supplementary Fig S1D.

Tl^+ leakage controls for the different added substrates. I was happy to see that the authors included a number of those (0 cAMP in Fig. 1, S2 and S4) for most substrates but they are never called out as such. Minor point as they all appear very flat but would be good to mention them explicitly and their affect (lack of) on the rate (maybe in SI).

We added a sentence on page 4: *Furthermore, neither WT SthK nor SthK P300A showed measurable activity in the absence of cAMP (grey lines in Figure 1D,E), similar to the quenching observed for protein-free liposomes (Supplementary Fig S1D).*

For the stopped-flow normalized rate numbers per sample, provided in the accompanying excel file. I was surprised so see a number of values as exactly 1 (and quite some even 1.0000). What is the reason for that?

The difference between 1 and 1.0000 originates from copying and pasting into an excel file from different source files. In both cases the number is simply 1, and we corrected this in the new excel file.

The reason for this comes from the normalization. Every single experiment was normalized separately in order to bring the separate, independent repeats to the same level for averaging.

We typically observe a variation of $\pm 10\%$ between experiments. We normalized the independent experiments to the highest rate constant of quenching (unless this was a clear outlier), which leads to the values of 1.

References:

- 1 Rheinberger, J., Gao, X., Schmidpeter, P. A. & Nimigean, C. M. Ligand discrimination and gating in cyclic nucleotide-gated ion channels from apo and partial agonist-bound cryo-EM structures. *Elife* **7**, doi:10.7554/eLife.39775 (2018).
- 2 Viswanadhan, V. N., Ghose, A. K., Revankar, G. R. & Robins, R. K. Atomic physicochemical parameters for three dimensional structure directed quantitative structure-activity relationships. 4. Additional parameters for hydrophobic and dispersive interactions and their application for an automated superposition of certain naturally occurring nucleoside antibiotics. *Journal of Chemical Information and Computer Sciences* **29**, 163-172, doi:10.1021/ci00063a006 (1989).
- 3 Hatton, G. I. & Yang, Q. Z. Ionotropic histamine receptors and H2 receptors modulate supraoptic oxytocin neuronal excitability and dye coupling. *J Neurosci* **21**, 2974-2982 (2001).
- 4 Di Benedetto, G., Scalzotto, E., Mongillo, M. & Pozzan, T. Mitochondrial Ca²⁺ uptake induces cyclic AMP generation in the matrix and modulates organelle ATP levels. *Cell Metab* **17**, 965-975, doi:10.1016/j.cmet.2013.05.003 (2013).
- 5 Firsov, A. M. *et al.* A conjugate of decyltriphenylphosphonium with plastoquinone can carry cyclic adenosine monophosphate, but not cyclic guanosine monophosphate, across artificial and natural membranes. *Biochim Biophys Acta Biomembr* **1860**, 329-334, doi:10.1016/j.bbamem.2017.10.013 (2018).
- 6 Peuker, S. *et al.* Kinetics of ligand-receptor interaction reveals an induced-fit mode of binding in a cyclic nucleotide-activated protein. *Biophys J* **104**, 63-74, doi:S0006-3495(12)05063-1 [pii]10.1016/j.bpj.2012.11.3816 (2013).
- 7 Goldschen-Ohm, M. P. *et al.* Structure and dynamics underlying elementary ligand binding events in human pacemaking channels. *Elife* **5**, doi:10.7554/eLife.20797 (2016).
- 8 Lolicato, M. *et al.* Tetramerization dynamics of C-terminal domain underlies isoform-specific cAMP gating in hyperpolarization-activated cyclic nucleotide-gated channels. *J Biol Chem* **286**, 44811-44820, doi:M111.297606 [pii]10.1074/jbc.M111.297606 (2011).

REVIEWERS' COMMENTS

Reviewer #3 (Remarks to the Author):

The authors have addressed our previous concerns

Reviewer #4 (Remarks to the Author):

I am quite satisfied with the updated version of this manuscript and I find that all my comments/question have been addressed. Overall, this study is well constructed and described, shedding significant light on how prolyl isomerases can influence channel gating.